# Inter-organ Wingless/Ror/Akt signaling regulates nutrient-dependent hyperarborization of somatosensory neurons

**Yasutetsu Kanaoka[1], Koun Onodera[1], Kaori Watanabe[1], Yusaku Hayashi[1], Tadao Usui[1], Tadashi Uemura[1,2,3]\*, Yukako Hattori[1,4]\***

[1]Graduate School of Biostudies, Kyoto University, Kyoto, Japan; [2]Research Center for Dynamic Living Systems, Kyoto University, Kyoto, Japan; [3]AMED-CREST, Tokyo, Japan; [4]JST FOREST, Tokyo, Japan

**\*For correspondence:**
tauemura@lif.kyoto-u.ac.jp (TU);
yhattori@lif.kyoto-u.ac.jp (YH)

**Competing interest:** The authors declare that no competing interests exist.

**Abstract** Nutrition in early life has profound effects on an organism, altering processes such as organogenesis. However, little is known about how specific nutrients affect neuronal development. Dendrites of class IV dendritic arborization neurons in *Drosophila* larvae become more complex when the larvae are reared on a low-yeast diet compared to a high-yeast diet. Our systematic search for key nutrients revealed that the neurons increase their dendritic terminal densities in response to a combined deficiency in vitamins, metal ions, and cholesterol. The deficiency of these nutrients upregulates Wingless in a closely located tissue, body wall muscle. Muscle-derived Wingless activates Akt in the neurons through the receptor tyrosine kinase Ror, which promotes the dendrite branching. In larval muscles, the expression of *wingless* is regulated not only in this key nutrient-dependent manner, but also by the JAK/STAT signaling pathway. Additionally, the low-yeast diet blunts neuronal light responsiveness and light avoidance behavior, which may help larvae optimize their survival strategies under low-nutritional conditions. Together, our studies illustrate how the availability of specific nutrients affects neuronal development through inter-organ signaling.

## Editor's evaluation

Nutrition profoundly affects neural development. The Uemura lab previously reported that C4da neurons elaborate complex dendrites when larvae grow on low-yeast diets, a phenomenon called neural sparing. In the current study, they define the molecular mechanism underlying the nutrition-mediated phenomenon and identify that the inter-organ Wingless/Ror/Akt pathway between the neuron and its adjacent muscles is necessary and sufficient to mediate dendrite over-branching in the low-yeast condition.

## Introduction

The physiological state of an organism influences organogenesis throughout the body. Among many external factors affecting the physiological state, nutrition in early life has a profound impact (***Bhutta et al., 2017***). This is particularly the case with neural development, which is highly metabolically demanding. A large amount of energy is consumed to control neural stem cell division, form complex dendrites and long axons in myriad neuronal cell types, and ultimately construct functional neural circuits (***Prado and Dewey, 2014***; ***Georgieff et al., 2015***). Compared to metabolic regulation of neural stem cell proliferation (***Homem et al., 2015***), little is known about how nutritional status is

conveyed to developing neurons and how those neurons regulate growth in response to such a signal (*Shimada-Niwa and Niwa, 2014*; *Shimono et al., 2014*; *Liu et al., 2017*).

Dietary nutrients are absorbed by the digestive tract and circulated throughout the body, and they are sensed by organs including the nervous system (*Chantranupong et al., 2015*). Those organs communicate the nutritional status to each other by secreting signaling molecules, either low-molecular-weight metabolites or macromolecules such as soluble proteins and lipoprotein particles, to elicit tissue-specific responses; and it is this inter-organ communication network that coordinates organogenesis with body growth (*Droujinine and Perrimon, 2016*; *Texada et al., 2020*). In the nervous system, neurons sense circulating nutrients directly or by way of signaling molecules derived from other tissues, so there exist diverse modes of nutrient sensing (*Morton et al., 2014*; *Jayakumar and Hasan, 2018*).

The above-mentioned regulatory mechanisms of nutrient-dependent neuronal development can be explored at the molecular level with appropriate model neurons; and one particularly amenable model is the *Drosophila* class IV dendritic arborization (C4da) neuron located in the larval periphery (*Grueber et al., 2002*). Dendritic arbors of C4da neurons extensively cover the body wall, and they are elaborated two-dimensionally between the epidermis and the body wall muscles. C4da neurons in larvae respond to noxious thermal, mechanical, and light stimuli and provoke robust avoidance behaviors (*Tracey et al., 2003*; *Hwang et al., 2007*; *Xiang et al., 2010*; *Zhong et al., 2010*; *Guntur et al., 2015*; *Chin and Tracey, 2017*). In the context of adaptation of growing animals to nutritional environments, it has been shown more recently that the C4da neurons sense amino acid deprivation by an amino acid transporter, Slimfast, at a late larval stage, which contributes to overcoming the nutritional stress, thereby allowing pupariation (*Jayakumar et al., 2016*; *Jayakumar et al., 2018*). In addition, we and another group have shown that dendrites of C4da neurons become more complex when larvae are reared on low-yeast diets compared to high-yeast diets (*Figure 1A and B*, and *Figure 1—figure supplement 1A*; *Watanabe et al., 2017*; *Poe et al., 2020*). We designate this counterintuitive phenotype as hyperarborization. Although the entire larval development takes longer on the low-yeast diet compared to the high-yeast diet (*Figure 1—figure supplement 1A*), it is unlikely that the hyperarborization is a simple consequence of the longer larval stage (*Watanabe et al., 2017*; see also Results). Therefore, it has been assumed that the low-yeast diet is deficient in select nutrients, which causes the phenotype. However, the identities of such key nutrients responsible for the hyperarborization phenotype have heretofore not been determined.

A wealth of genetic analyses on standard foods has revealed numerous regulators of dendrite morphogenesis working either in cell-autonomous or non-cell autonomous manners (*Jan and Jan, 2010*; *Dong et al., 2015*; *Valnegri et al., 2015*). Some of the cell-autonomous mechanisms include those related to intake and synthesis of metabolites: amino acid transporter SLC36/Pathetic (Path) (*Lin et al., 2015*) and a critical regulator of fatty acid synthesis, sterol regulatory element binding protein (SREBP) (*Meltzer et al., 2017*; *Ziegler et al., 2017*). Concerning the non-cell autonomous mechanisms, direct interactions between C4da neurons and one of the adjacent tissues, the epidermis, have been well characterized with the help of anatomical approaches under both light and electron microscopes (*Yang and Chien, 2019*). Some portions of dendritic branches are attached to the extracellular matrix, and the attachment is mediated by signaling between an epidermally derived semaphorin ligand Sema-2b and its receptor Plexin B (PlexB) on the dendrite (*Meltzer et al., 2016*), as well as between a TGF-β ligand Maverick (Mav)-Ret receptor combination (*Hoyer et al., 2018*). Other portions of dendritic arbors are wrapped by epidermal cells, so overall the dendrite arbor is embedded in the epidermis locally (*Han et al., 2012*; *Kim et al., 2012*; *Tenenbaum et al., 2017*; *Jiang et al., 2019*). In contrast to the above dendrite-epidermis interaction, there is much less evidence for signaling between muscles and dendrites, despite their proximity to dendrites and their large volume in the body (*Yasunaga et al., 2010*). Furthermore, when considering the relationship between nutritional status and C4da neurons, little is known about how exactly the dietary information is remotely transmitted from the gut to the neurons. To address these unsolved questions, it is critical to efficiently quantify the effects of various nutritional and genetic conditions on this nutrition-dependent hyperarborization. For this purpose, we developed DeTerm, a software program for automatic detection of dendritic branch terminals (*Figure 1A and B*; *Kanaoka et al., 2019*).

Here, we show that C4da neurons increase their dendritic terminal density on a low-yeast diet (LYD) compared to a high-yeast diet (HYD) due to a concurrent deficiency in vitamins, metal ions, and

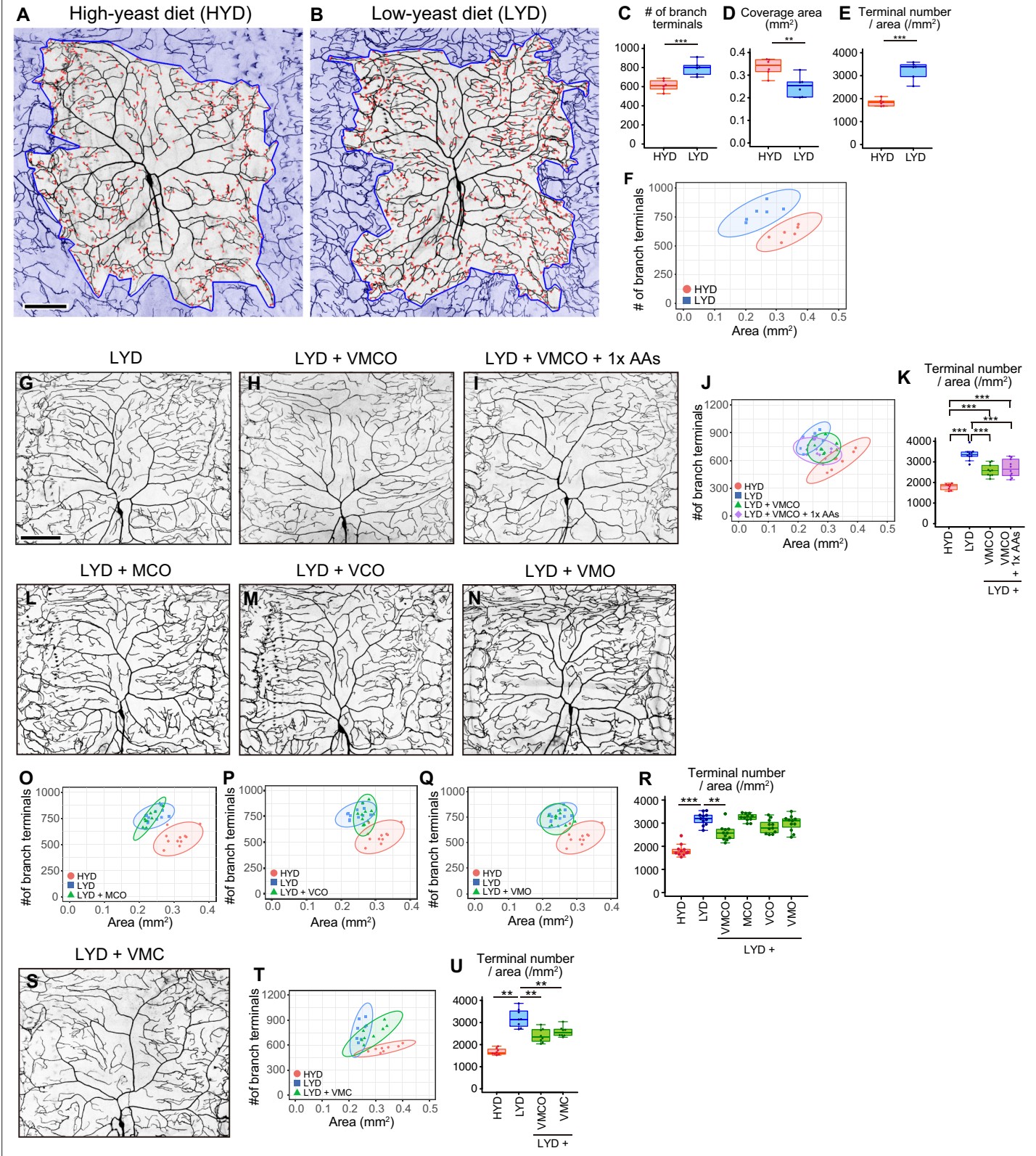

**Figure 1.** A mixture of vitamins, metal ions, and cholesterol ameliorates C4da neuron hyperarborization. (**A and B**) Representative output images of DeTerm. DeTerm automatically detects dendritic terminals of C4da neuron ddaC in larvae reared on a high-yeast diet (HYD; **A**) or a low-yeast diet (LYD; **B**). Red points indicate detected branch terminals. (**C–E**) The numbers of branch terminals detected by DeTerm (**C**), coverage areas of dendrites (**D**), and densities of branch terminals (the number of branch terminals/coverage area; **E**) of individual neurons, on HYD or LYD (Student's t-test, n=6). Boxes show

*Figure 1 continued on next page*

*Figure 1 continued*

the 25th–75th percentiles. The central lines indicate the medians. Whiskers extend to the most extreme data points, which are no more than 1.5 times the interquartile range. Boxes and points for HYD data and those for LYD data are colored red and blue, respectively, in this and subsequent figures. (**F**) Two-dimensional (2D) plot of the dendritic area and the number of branch terminals of each neuron. The ellipses represent the 95% confidence intervals, which are clearly separated for HYD and LYD. This plot shows a positive correlation between the area of the dendritic field and the number of branch terminals. (**G–I**) Images of ddaC neurons in larvae reared on LYD (**G**), LYD +vitamin + metal ion +cholesterol + other ingredients (LYD +VMCO; **H**), or LYD +VMCO + 1 x amino acids (LYD +VMCO + 1 x AAs; **I**). (**J and K**) Quantitative analysis of ddaC on LYD +VMCO or LYD +VMCO + 1 x AAs. (**J**) 2D plot. Note that the ellipse of LYD +VMCO and that of LYD +VMCO + 1 x AAs are located between those of HYD and LYD. (**K**) Densities of branch terminals (One-way ANOVA and Tukey's HSD test, n=8–10). (**L–N**) Images of ddaC neurons in larvae reared on LYD +metal ion+cholesterol + other ingredients (LYD +MCO; **L**), LYD +vitamin + cholesterol +other ingredients (LYD +VCO; **M**), or LYD +vitamin + metal ion +other ingredients (LYD +VMO; **N**). (**O–R**) 2D plots of ddaC on LYD +MCO (**O**), LYD +VCO (**P**), or LYD +VMO (**Q**), and densities of branch terminals (R, Steel test, n=10). The elipses of these diets largely overlap with that of LYD and clearly or almost separare from that of HYD (**O–Q**). (**S**) Images of ddaC neurons in larvae reared on LYD +vitamin + metal ion +cholesterol (LYD +VMC). (**T and U**) 2D plots of ddaC neurons in larvae reared on LYD +VMC (**T**) and densities of branch terminals (U, Steel test, n=8). The ellipse of LYD +VMC is located between those of HYD and LYD (**T**). Boxplots in (**K, R and U**) are depicted as in (**C**). *p<0.05, **p<0.01, and ***p<0.001. Scale bars, 100 µm.

The online version of this article includes the following figure supplement(s) for figure 1:

**Figure supplement 1.** Addition of amino acids does not rescue the hyperarborization.

**Figure supplement 2.** Extended larval growth is not associated with the hyperarborization.

cholesterol. We then identified an extrinsic factor and an intracellular signaling axis that jointly enable C4da neurons to respond to the LYD nutritional status. On LYD, Akt and its upstream receptor tyrosine kinase Ror in the neuron are required for the hyperarborization. In a paracrine fashion, Wingless (Wg) produced by body wall muscles activates Akt by way of Ror and contributes to the hyperarborization. In muscles of larvae on the HYD, Stat92E, a transcription factor in the JAK/STAT pathway, was more highly expressed and negatively regulated *wg* expression, whereas the LYD resulted in lower expression of *Stat92E*, which partly contributed to higher expression of *wg*. Together, our studies illustrate how nutritional environments impact neuronal development through the Wg-Ror-Akt pathway between the neuron and closely located muscles. As for the neuronal function, we found that LYD blunted light responsiveness of class IV neurons and larval light avoidance behavior, which may help larvae optimize their survival strategies under low-nutritional conditions.

## Results

### A mixture of vitamins, metal ions, and cholesterol ameliorates the hyperarborization

Our analysis using the software program called DeTerm established that both the number of branch terminals per neuron and the density of terminals (terminal number/arbor size) were higher on LYD than on HYD (*Figure 1C-E*). In addition to these box plots, we drew two-dimensional plots with the dendritic area on the X-axis and the number of branch terminals on the Y-axis, and the numerical features of dendrites of C4da neurons on HYD and those on LYD were clearly separated (*Figure 1F*). Therefore, in the subsequent analyses, we mainly focused on the density of branch terminals (*Figure 1E*) and the separation in 2D plots (*Figure 1F*) to evaluate the hyperarborization phenotypes.

Yeast is one of the main ingredients in *Drosophila* laboratory foods, and it has been primarily considered as a source of amino acids. We suspected the possibility that LYD is deficient in amino acids and that is the cause of the phenotype. Therefore, we first examined whether supplementation of LYD with amino acids would ameliorate the hyperarborization. However, the addition of an essential amino acid solution, an amino acid mix, or peptone resulted in only slight or no restoration of the phenotype (*Figure 1—figure supplement 1B–L*, and see details in the legend). To more comprehensively search for nutrients responsible for the hyperarborization, we used fractions of a fully chemically defined or holidic medium for *Drosophila* (*Piper et al., 2014*; *Piper et al., 2017*) and examined which fraction or which combinations of the fractions were able to ameliorate the phenotype (*Figure 1—figure supplement 1A*). Addition of four fractions other than amino acids, which comprise vitamins (V), metal ions (M), cholesterol (C), and other ingredients (nucleic acids and lipid-related metabolites: O), to LYD significantly rescued the hyperarborization (*Figure 1G, H, J and K*; see also the legend of *Figure 1J*). We named this diet LYD + VMCO. Further supplementation of amino acids to LYD + VMCO did not

improve the degree of the rescue (*Figure 1K*). Importantly, the phenotype was not restored without any one of three fractions, namely, vitamins, metal ions and cholesterol (*Figure 1L–R*). On the other hand, the fraction designated other ingredients was dispensable for amelioration of the phenotype (*Figure 1S-U*). These results suggest that the concurrent deficiency in vitamins, metal ions, and cholesterol contributes to the hyperarborization phenotype.

Larval developmental timing on LYD + VMC was comparable to that on LYD *Figure 1—figure supplement 1A*; nonetheless the hyperarborization phenotype was blunted on LYD + VMC. We then examined whether extension of larval growth cause the hyperarborization by testing other dietary or genetic interventions. We previously compared dendrite morphologies between larvae reared on a low-sugar diet and those on a high-sugar diet that delays larval development (*Musselman et al., 2011*), and we reported that the hyperarborization does not occur on the high-sugar diet (*Watanabe et al., 2017*). We expanded this approach and analyzed the effect of the sugar overload on dendrite branching in a quantitative manner. When we observed dendrites in larvae reared on HYD supplemented with sucrose at the same timing as those on LYD (8–9 days AEL), they did not become more complex compared to those on HYD alone (*Figure 1—figure supplement 2A–E*). Moreover, we observed the C4da neurons in larvae with *dlip8* overexpressed in wing imaginal discs, which is sufficient to extend the larval stage (*Colombani et al., 2012*). This genetic intervention did not affect dendrite complexity (*Figure 1—figure supplement 2F–J*). Altogether, these results suggest that the prolonged larval period was not the primary cause of the hyperarborization phenotype.

## Akt and receptor tyrosine kinase Ror are required in C4da neurons to hyperarborize their dendrites

To investigate the molecular mechanism underlying the hyperarborization phenotype on LYD, we focused on intracellular signaling factors that have been reported to sense nutritional status in other cellular contexts. Thus, we examined whether C4da neuron-specific knockdown (KD) of any of these factors would affect this diet-selective phenotype (*Figure 2A–J* and *Figure 2—figure supplement 1A–P*). To identify candidate genes whose KD attenuated the hyperarborization phenotype, we investigated how much the HYD ellipse and the LYD ellipse approached each other or overlapped in the 2D plot. We also compared the terminal density using two-way analysis of variance (ANOVA) throughout this study, unless described otherwise (see p-values in the aforementioned figures).

Among KD phenotypes of the candidate genes, we were interested in an *Akt kinase* (*Akt*) KD in one of the RNAi lines, which impacted hyperarborization but left overall dendritic architecture relatively intact (v2902; *Figure 2B, E, G and H*). This *Akt* KD in the v2902 line resulted in neither apparent downsizing of the arbor area (*Figure 2G*) nor overt decreases in branch length on HYD (*Figure 2—figure supplement 2A and B*), contrasting with diet-independent severe phenotypes observed in another *Akt* RNAi line (BL33615; *Figure 2C, F, I and J* and *Figure 2—figure supplement 2C and D*). Our subsequent analyses showed that v2902 was less effective in knocking down *Akt* than BL33615 (*Figure 2—figure supplement 3A–H*). We interpreted these results as follows: the severe reduction of Akt function in the BL33615 line impairs growth of dendritic branches, as shown previously (*Parrish et al., 2009*), whereas the mild reduction in the v2902 line mostly secures the basal activity of Akt necessary for growth, but it affects hyperarborization on LYD in a relatively selective manner (*Figure 2—figure supplement 2G–I*). We further knocked down genes that constitute the signaling pathways of Akt (*Figure 2—figure supplement 1Q-AJ*), and found that inhibition of TOR signaling components, such as *Target of rapamycin* (*Tor*) or *Ribosomal protein S6 kinase* (*S6k*) also ameliorated the phenotype (*Figure 2—figure supplement 1Q–AJ*). A similar effect caused by inhibition of *Tor* was described by *Poe et al., 2020*.

Various secreted factors are known to function as inter-organ communication factors in response to nutritional conditions (*Droujinine and Perrimon, 2016*). We therefore hypothesized that, in larvae on LYD, C4da neurons receive signaling molecules from other tissues, leading to the hyperarborization via the Akt/Tor signaling pathway. As candidate receptors upstream of Akt, we focused on receptor tyrosine kinases (RTKs; *Sopko and Perrimon, 2013*), and conducted C4da neuron-specific KD screenings of 20 RTK genes (*Supplementary file 2* and *Figure 2—figure supplements 4–6*; see 'RTK screening' in Materials and methods). One of the positive hits in our primary screening was *RTK-like orphan receptor* (*Ror*; *Figure 2—figure supplement 4C and D*) and we confirmed that the *Ror* KD significantly suppressed the hyperarborization in the secondary screening (*Figure 2K–P*).

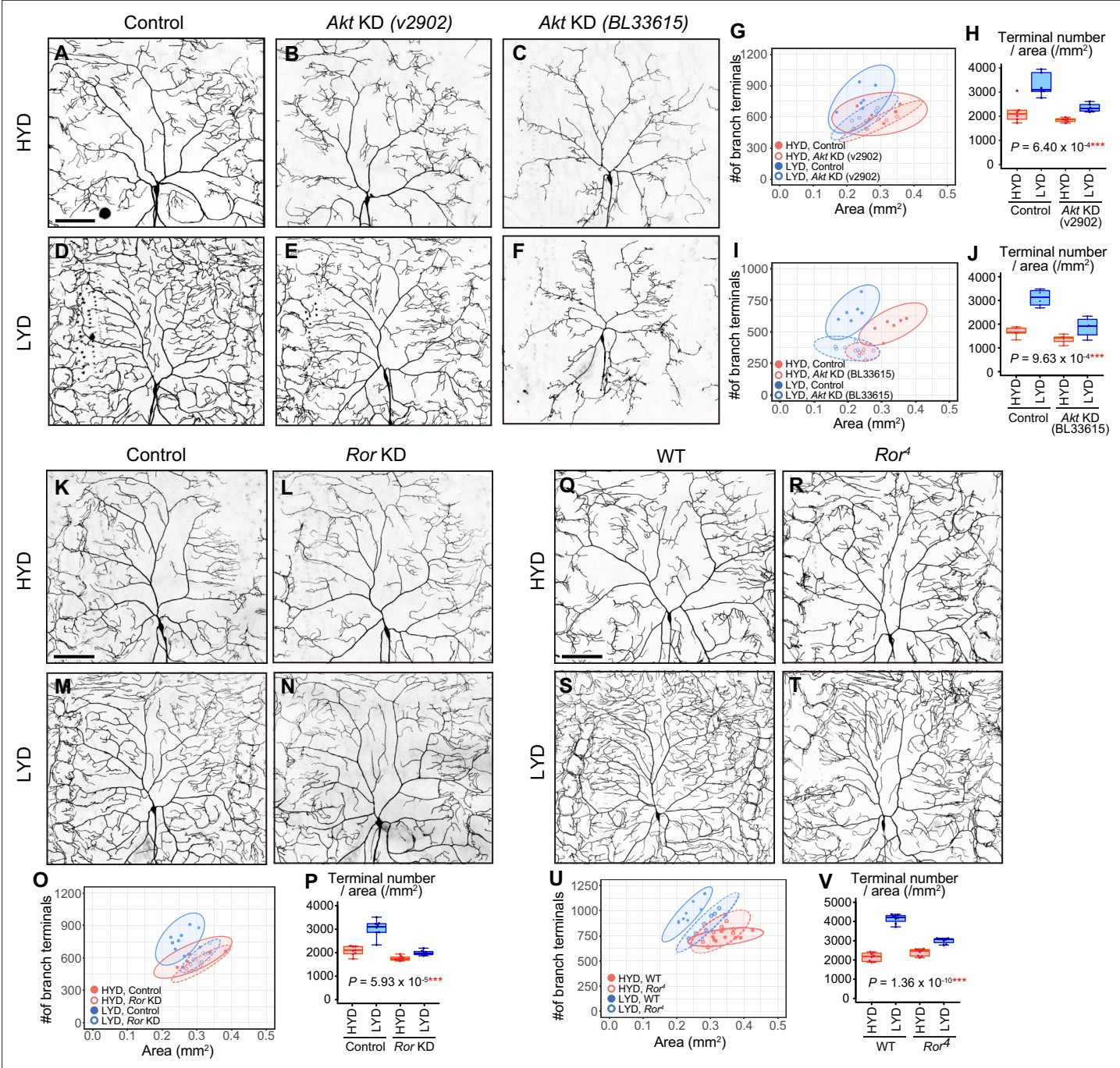

**Figure 2.** Akt and receptor tyrosine kinase Ror are required in C4da neurons to hyperarborize their dendrites. (**A–F**) Images of control ddaC neurons (**A and D**), or *Akt* knockdown (KD) ddaC neurons using *UAS-Akt RNAi^{v2902}* (**B and E**) or *UAS-Akt RNAi^{BL33615}* (**C and F**), on HYD (**A–C**) or LYD (**D–F**). (**G–J**) Quantitative analysis of effects of *Akt* KD using *UAS-Akt RNAi^{v2902}* (**G and H**) or *UAS-Akt RNAi^{BL33615}* (**I and J**). (**G and I**) 2D plots. (**H and J**) Densities of branch terminals. As indicated by the p-values, there was a significant interaction between diet and genotype on terminal density in both *Akt* KD experiments (two-way ANOVA, n=6). That is, compared to the difference between HYD and LYD in control C4da neurons, that difference in *Akt* KD neurons using *UAS-Akt RNAi^{v2902}* or *UAS-Akt RNAi^{BL33615}* was significantly smaller, suggesting that the hyperarborization was blunted by the *Akt* KD. (**K–P**) Images of control (**K and M**) or *Ror* knocked down ddaC (**L and N**) on HYD (**K and L**) or LYD (**M and N**). 2D plot (**O**) and densities of branch terminals (p, two-way ANOVA, n=8). (**Q–V**) Images of ddaC in wild-type (WT; **Q and S**) or *Ror^4* mutant larvae (**R and T**) on HYD (**Q and R**) or LYD (**S and T**). The ddaC neurons were visualized by expressing *ppk-CD4:tdGFP*. 2D plot (**U**) and densities of branch terminals (V, two-way ANOVA, n=8). Boxplots in (H, J, P, and V) are depicted as in *Figure 1C*. ***p<0.001. Scale bars, 100 µm.

The online version of this article includes the following figure supplement(s) for figure 2:

*Figure 2 continued on next page*

**Figure supplement 1.** Contributions of intracellular signaling factors or Akt signaling components to the hyperarborization.

**Figure supplement 2.** Roles of Akt and Ror in dendritic branching and elongation.

**Figure supplement 3.** Validation of knockdown efficiency of *Akt* RNAi lines and p-Akt antibody specificity.

**Figure supplement 4.** Results of the primary knockdown screening of RTK genes.

**Figure supplement 5.** Knockdown effects of *InR, Alk* or *htl* in the primary and the secondary screenings.

**Figure supplement 6.** RTK genes that were negative in the secondary screening.

Moreover, we observed C4da neurons in *Ror*[4] null mutant larvae and showed that they recapitulated the result of the KD (*Figure 2Q–V*). These results suggest that Ror and Akt are required in C4da neurons to hyperarborize their dendrites on LYD. Other positive hits included the known upstream regulators of Akt, Insulin-like receptor (InR) or Anaplastic lymphoma kinase (Alk). However, we could not definitively conclude whether InR and Alk contribute to the hyperarborization phenotype, due to inconsistent KD results among multiple experiments (*Supplementary file 2* and *Figure 2—figure supplement 5*).

To further characterize the phenotypes of the *Ror* KD and the v2902 *Akt* KD line, we measured the total length of branches per neuron (dendrite length) and dendrite length/area on each diet (*Figure 2—figure supplement 2A, B, E, F*). For both KD lines, values for length were higher on LYD compared to HYD in control C4da neurons. The v2902 line reduced dendrite length on both diets, although the decrease on HYD was marginal (*Figure 2—figure supplement 2A and B*), whereas *Ror* KD only decreased dendrite length on LYD (*Figure 2—figure supplement 2E and F*). This difference may reflect a restricted role for Ror in the response to a deficiency in the key nutrients, as opposed to a more general requirement of Akt for branch growth irrespective of the diets (*Figure 2—figure supplement 2G, H, J*).

## Wg in muscles is more highly expressed on LYD and promotes dendritic branching of C4da neurons

Ror binds to Wnt ligands and triggers intracellular signaling cascades (*Ripp et al., 2018*; *van Amerongen and Nusse, 2009*). We therefore knocked down *wingless* (*wg*), *Wnt2*, *Wnt4*, or *Wnt5* in either of the two tissues adjacent to C4da neurons: epidermal cells and muscles. We observed that *wg* KD in muscles using either *Mhc-GAL4* or *mef2-GAL4* suppressed the hyperarborization phenotype (*Figure 3A–F* and *Figure 3—figure supplement 1A–F*). In contrast, epidermal KD of *wg* had no effect on the phenotype (*Figure 3—figure supplement 1G–L*). The requirement of Wg for the hyperarborization was further confirmed by the finding that the hyperarborization effect was dampened in C4da neurons in the whole-body *wg* mutant (hypomorphic *wg*[1]/amorphic *wg*[l-8]; *Figure 3G–L*).

We then examined whether Wg is differentially expressed in muscles between larvae reared on HYD and those on LYD. Immunostaining using an anti-Wg antibody showed stronger signals in LYD-fed larvae (*Figure 3M, N and P*). These stronger signals indeed represented increased amounts of endogenous Wg because knocking down *wg* decreased the intensity (*Figure 3—figure supplement 1M–Q*). We also asked whether *wg* expression is up-regulated on LYD at the transcriptional level. We expressed RedStinger, DsRed tagged with a nuclear localization signal, under the knocked-in *wg-GAL4* driver that reflects the endogenous expression pattern of *wg* (*Bosch et al., 2020*). Nuclear RedStinger signals in muscles were stronger in larvae on LYD (*Figure 3Q-S*), indicating that LYD up-regulated *wg* transcription compared to HYD. We further tested whether muscle-derived Wg promotes dendritic branching of C4da neurons. For this purpose, we overexpressed *wg* in muscles and found that those larvae increased the number of dendritic terminals per neuron on both HYD and LYD (*Figure 3T–Y*), strengthening the role of the muscle–C4da neuron communication in hyperarborization. Importantly, addition of vitamins, metal ions, and cholesterol to LYD significantly suppressed the up-regulation of Wg on LYD (*Figure 3M-P*). Together with the effect of these compounds on dendritic branching (*Figure 1S-U*), we hypothesized that *wg* expression in muscles is enhanced by a concurrent deficiency in vitamins, metal ions, and cholesterol in LYD, and that muscle-derived Wg promotes dendritic branching of C4da neurons.

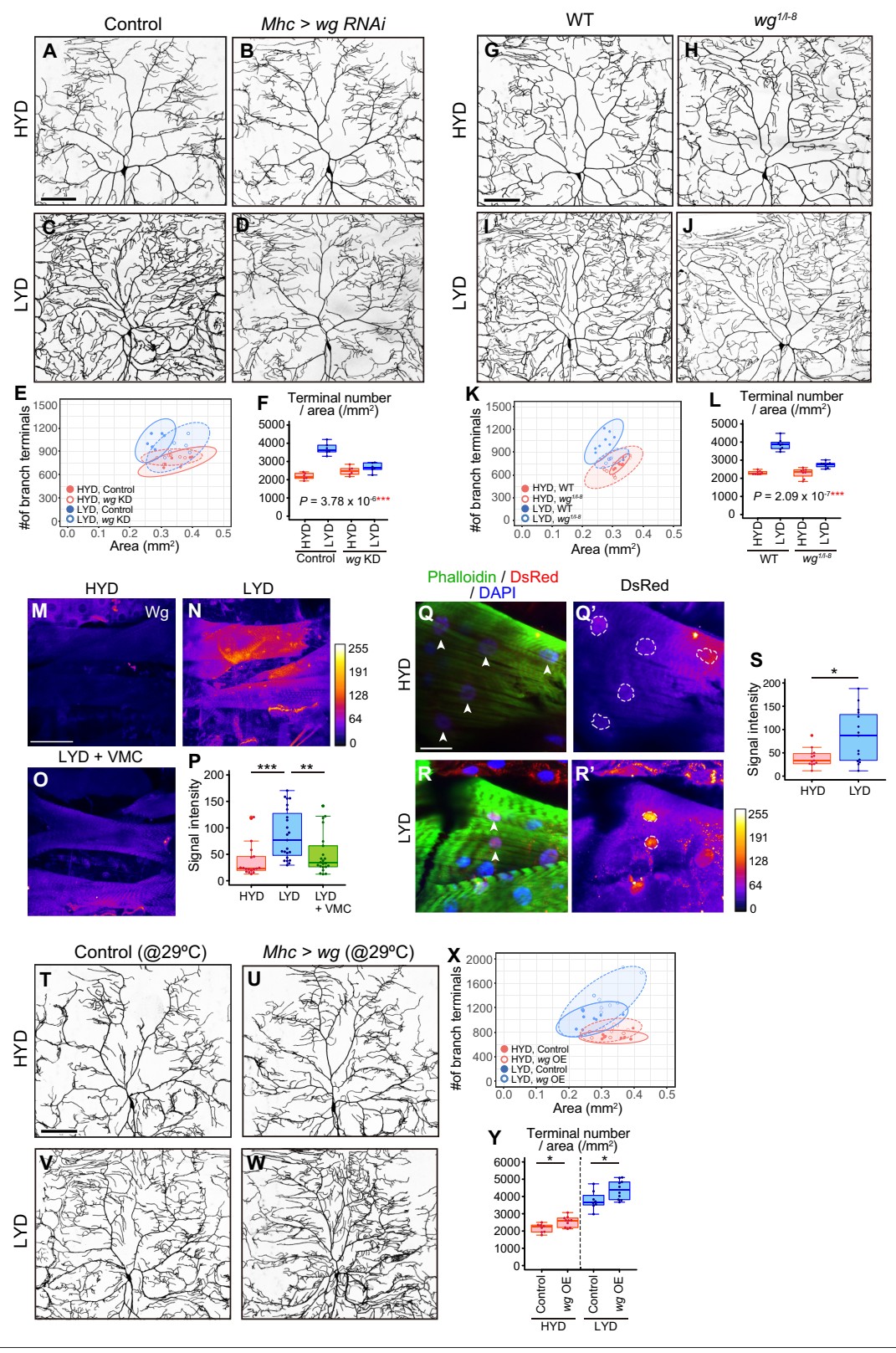

**Figure 3.** Wg in muscles is expressed more highly on LYD and promotes dendritic branching of C4da neurons. (**A–F**) Images of ddaC neurons in control larvae (**A and C**) or larvae with *wg* KD in muscles (**B and D**), on HYD or LYD. 2D plot (**E**) and densities of branch terminals (F, two-way ANOVA, n=6). (**G–L**) Images of ddaC neurons in WT (**G and I**) or *wg^1/I-8* larvae (**H and J**) on HYD or LYD. 2D plot (**K**) and densities of branch terminals (L, two-way ANOVA,

*Figure 3 continued on next page*

*Figure 3 continued*

n=8). (**M–P**) Muscles in larvae reared on HYD (**M**), LYD (**N**), or LYD +VMC (**O**) were stained for Wg. The signal intensities are represented by the indicated color code. (**P**) Quantification of the mean Wg immunofluorescence intensity in muscle 9, one of the closest muscles to the ddaC neuron (Steel Dwass test, n=18–23). (**Q–S**) Images of *wg-Gal4* muscles driving the expression of RedStinger, which is DsRed tagged with a nuclear localization signal. Muscles, on HYD (**Q**) or LYD (**R**) were stained with phalloidin (green), an antibody to DsRed (red), and DAPI (blue). The signal intensities of DsRed are represented by the indicated color code, and white dashed circles indicate outlines of nuclei (Q'and R'). (**S**) Quantification of DsRed intensity in nuclei of muscle 9 (Wilcoxon-Mann-Whitney test, n=13–15). (**T–Y**) Images of ddaC neurons in control larvae (**T and V**) or larvae with *wg* overexpression at 29 °C in muscles (**U and W**) on HYD or LYD. 2D plot (**X**) and densities of branch terminals (Y, Wilcoxon-Mann-Whitney test, n=8–10). Experiments were couducted at both 29°C and 25°C, but only results at 29 °C are shown. At 29 °C, the effect of *wg* overexpression was expected to be higher, and in fact, branch terminal density increased on both HYD and on LYD, but only on HYD at 25 °C. The increase in the branch terminal density elicited by *wg* overexpression on both diets was less dramatic than the difference in the branch terminal density due to the diets in each respective genotype. The ddaC neurons were visualized by expressing *ppk-CD4:tdGFP*. Boxplots in (F, L, P, S, and Y) are depicted as in *Figure 1C*. \*p<0.05, \*\*p<0.01, and \*\*\*p<0.001. Scale bars, 100 μm (A-D, G-J, M-O, and T-W), 25 μm (**Q-R'**).

The online version of this article includes the following figure supplement(s) for figure 3:

**Figure supplement 1.** Wg from muscles, but not from epidermal cells, contributes to the hyperarboriztion phenotype.

## Wg-Ror-mediated activation of Akt in C4da neurons evokes the hyperarborization

Wnt signaling is engaged in diverse contexts of neuronal development and regeneration (*Green et al., 2014*; *He et al., 2018*; *Endo and Minami, 2018*; *Nye et al., 2020*; *Weiner et al., 2020*). In a previous study in *Drosophila*, responses to dendrite injuries were investigated using class I da (C1da) and C4da neurons. This showed that Ror, a seven-pass transmembrane receptor Frizzled (Fz), and downstream components including Disheveled (Dsh) and Axin (Axn) are required for dendrite regeneration (*Nye et al., 2020*). We therefore examined whether these genes and other components of Wnt signaling affect the hyperarborization phenotype (*Figure 4—figure supplement 1* and *Figure 4—figure supplement 2*). Knocking down *fz2* significantly ameliorated the hyperarborization (*Figure 4—figure supplement 1A, C, D, F, H, J*). Not only *fz2* KD neurons, but also *fz2* null mutant neurons showed less prominent hyperarborization compared to the control neurons (*Figure 4—figure supplement 1K–P*). These results are consistent with the proposed function of Ror as a Wnt co-receptor with Fz2 (*Ripp et al., 2018*). In addition to *fz2*, KD of *fz*, KD of downstream components (*dsh* and *Axn*), or expression of a dominant-negative form of Bsk also significantly blunted the hyperarborization (*Figure 4—figure supplement 1B, E, G, I* and *Figure 4—figure supplement 2* [B, I, O and U], [G, N, T and Z], and [C, J, P and V]). However, we question whether all of these ameliorated phenotypes share the same underlying mechanism with those of *Ror* or *fz2* KD (see the legend of *Figure 4—figure supplements 1 and 2*, and DISCUSSION). Altogether, our results suggest that among the known components of Wnt signaling in *Drosophila*, at least Fz2 cooperates with Ror in transducing the external signal to evoke the hyperarborization.

Ror is also reported to activate the PI3K/Akt/mTor signaling pathway in lung adenocarcinoma or multiple myeloma (*Liu et al., 2015*; *Frenquelli et al., 2020*). We therefore hypothesized that Wg-Ror signaling activates Akt signaling in C4da neurons on LYD, leading to the hyperarborization. To test this hypothesis and to clarify the relationship between Wg-Ror and Akt at the molecular level, we examined how genetic manipulations of Wg-Ror signaling affect Akt activity levels in C4da neurons (*Figure 4A–J*). The specificity of the p-Akt antibody in C4da neurons was validated in two ways: (1) p-Akt signals were significantly reduced by *Akt* KD *Figure 2—figure supplement 3A-H*; (2) expression of myr-Akt, a constitutively activated membrane-anchored form of Akt (*Stocker et al., 2002*), dramatically increased the p-Akt signal strength (*Figure 2—figure supplement 3I and J*). Using this antibody, we first examined how the p-Akt level in C4da neurons differs between larvae reared on HYD and LYD. Immunostaining showed that the p-Akt level in C4da neurons was higher on LYD than on HYD (*Figure 4A, A', C, C' , and E*). In contrast, *Ror* KD neurons from larvae on LYD showed reduced p-Akt levels compared to those on HYD (*Figure 4B, B', D, D' , and E*). Furthermore, *wg* overexpression

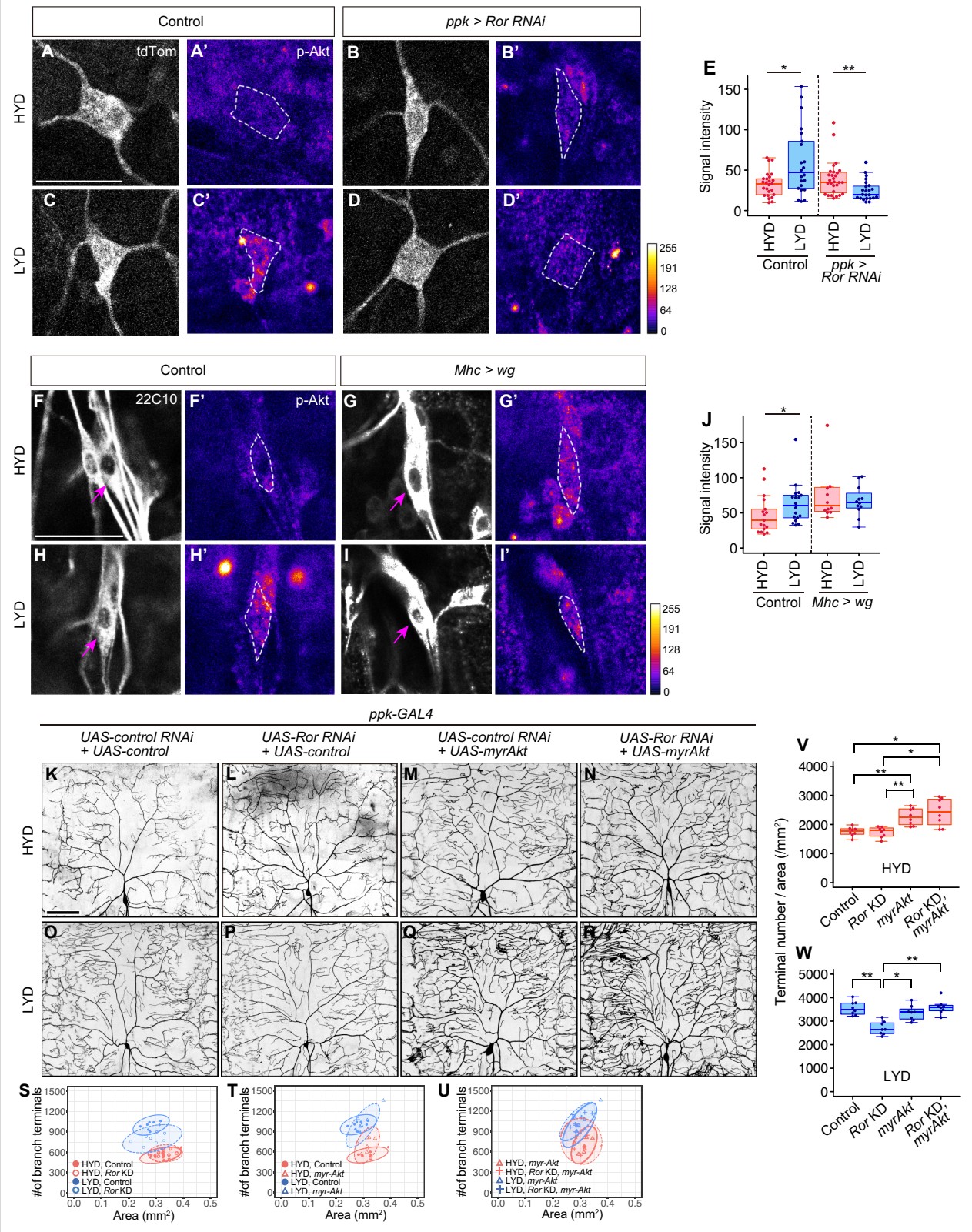

**Figure 4.** Wg-Ror-mediated activation of Akt in C4da neurons evokes the hyperarborization. (**A–E**) Control ddaC (**A, A', C and C'**) or *Ror* KD ddaC neurons (**B, B', D and D'**) were stained for p-Akt (**A'-D'**) and co-imaged with a C4da neuron marker *ppk-CD4:tdTom* (**A–D**). The signal intensities of p-Akt correspond to the indicated color code, at right, and white dashed circles indicate the cell bodies of ddaC neurons. (**E**) Quantification of p-Akt intensity in cell bodies of control or *Ror* knocked down ddaC neurons (Wilcoxon-Mann-Whitney test, n=22–28). (**F–J**) ddaC in control larvae (**F, F', H and H'**) or

*Figure 4 continued on next page*

*Figure 4 continued*

larvae with *wg* overexpression in muscles (**G, G′, I and I′**) were stained for a pan-sensory neuron marker (22C10; **F–I**) and for p-Akt (**F′-I′**). (**F–I**) Magenta arrows indicate the cell bodies of ddaC neurons. (**F′-I′**) The intensities of p-Akt signals correspond to the indicated color code, at right, and white dashed circles indicate the cell bodies of ddaC neurons. (**J**) Quantification of p-Akt intensity in control larvae or larvae with *wg* overexpression in muscles (Wilcoxon-Mann-Whitney test, n=12–18). (**K–U**) Images of *UAS-control RNAi* and *UAS-control* expressing ddaC neurons (**K and O**), *UAS-Ror RNAi* and *UAS-control* expressing ddaC neurons (**L and P**), *UAS-control RNAi* and *UAS-myrAkt*, a constitutively active form of Akt, expressing ddaC neurons (**M and Q**), or *UAS-Ror RNAi* and *UAS-myrAkt* expressing ddaC neurons (**N and R**), on HYD or LYD. We used *UAS-grnd RNAi,* which had no significant impact on the hyperarborization phenotype (*Figure 2—figure supplement 1A and E*, 1I, and 1 M), as the *UAS-control RNAi* and *UAS-CD4:tdTom* as the *UAS-control*. (**S–W**) Quantitative analysis of combinatorial effects of *Ror* KD and *myrAkt* expression. (**S–U**) 2D plots. (**V and W**) Densities of branch terminals on HYD (**V**) or LYD (**W**) (Steel Dwass test, n=8–9). Boxplots in (**E, J, V and W**) are depicted as in *Figure 1C*. *p<0.05 and **p<0.01. Scale bars, 25 μm (**A-D′ and F-I′**), 100 μm (**K–R**).

The online version of this article includes the following figure supplement(s) for figure 4:

**Figure supplement 1.** Fz2, a receptor for Wnt proteins, is required in C4da neurons to hyperarborize their dendrites.

**Figure supplement 2.** Effects of inhibiting intracellular Wnt signaling components on hyperarborization.

**Figure supplement 3.** C3da neurons increased the number of dendrite terminals and p-Akt levels on LYD, while C1da neurons did not.

in muscles increased the p-Akt level in C4da neurons on HYD (compare *Figure 4F′* with 4 G′), which became comparable to the level on LYD (compare *Figure 4G′* with *Figure 4I′*; see quantification in *Figure 4J*). These results suggest that Akt signaling in C4da neurons is activated by muscle-derived Wg, and this activation is mediated by Ror in the neurons.

We further examined whether the activation of Akt itself evokes hyperarborization even in the absence of the upstream Ror-mediated signaling (*Figure 4K–W*). Expression of myr-Akt in C4da neurons increased the terminal density even on HYD, regardless of whether *Ror* was knocked down or not (compare *Figure 4K* with 4 M and 4 N; see also 4T, 4 U and 4 V). This result suggests that Akt activation in the neurons plays a pivotal role for the hyperarborization. Our result is consistent with a previous finding that overexpression of the wild-type form of *Akt* causes a significant increase in dendrite coverage of the epidermis (*Parrish et al., 2009*).

Somatosensation of *Drosophila* larvae depends on C4da and 3 other classes of da neurons. Among them, class I da (C1da) and class III da (C3da) neurons function in proprioception and gentle-touch sensation, respectively (*Hughes and Thomas, 2007*; *Hwang et al., 2007*; *Im and Galko, 2012*; *Yan et al., 2013*; *Tsubouchi et al., 2012*). We also examined whether the hyperarborization phenotype and p-Akt upregulation are observed in these classes. We previously reported that the hyperarborization phenotype is not seen in ddaD and ddaE C1da neurons (*Watanabe et al., 2017*). The p-Akt level showed no significant difference between HYD and LYD in ddaE neurons (*Figure 4—figure supplement 3A–C*). On the other hand, a C3da neuron, ddaF, showed an increase in both the dendritic terminal number and the p-Akt level on LYD (*Figure 4—figure supplement 3D–I*), similar to C4da neurons. Increased branch terminals of C3da neurons on a low-yeast diet was also reported previously (*Poe et al., 2020*). Our results raise the possibility that, along with C4da neurons, C3da neurons share the Akt-driven branching mechanism in response to the low-nutrient condition.

## Stat92E partially contributes to downregulation of Wg expression and suppresses hyperarborization on HYD

Given that Wg expression in muscles is higher on LYD (*Figure 3M–P*), and the differential expression impacts the dendritic branching of C4da neurons (*Figure 3T–Y*), we then asked how Wg expression in muscles is regulated in the nutrient-dependent manner. To search for upstream regulators of the Wg expression, we performed RNA-seq analysis on mature whole larvae that were reared on either diet. We identified 3854 differentially expressed genes between the diets (*Figure 5A*, *Figure 5—figure supplement 1*, and *Supplementary file 3*). Among these, we focused on a transcriptional factor in the JAK/STAT pathway, *Stat92E*, which is more highly expressed on HYD than LYD (*Figure 5B*). Also informing our decision, it was reported that Stat92E is a negative regulator of *wg* expression in the eye imaginal disc (*Ekas et al., 2006*). We used a Stat92E reporter strain (*Bach et al., 2007*) and found that Stat92E reporter expression in muscle was higher on HYD (*Figure 5C–E*). We therefore hypothesized that, in muscles of larvae on HYD, higher expression of Stat92E downregulates Wg expression, thereby suppressing the hyperarborization phenotype. To test this hypothesis, we knocked down *Stat92E* in muscles, and this led to increased Wg levels compared to the control muscles on

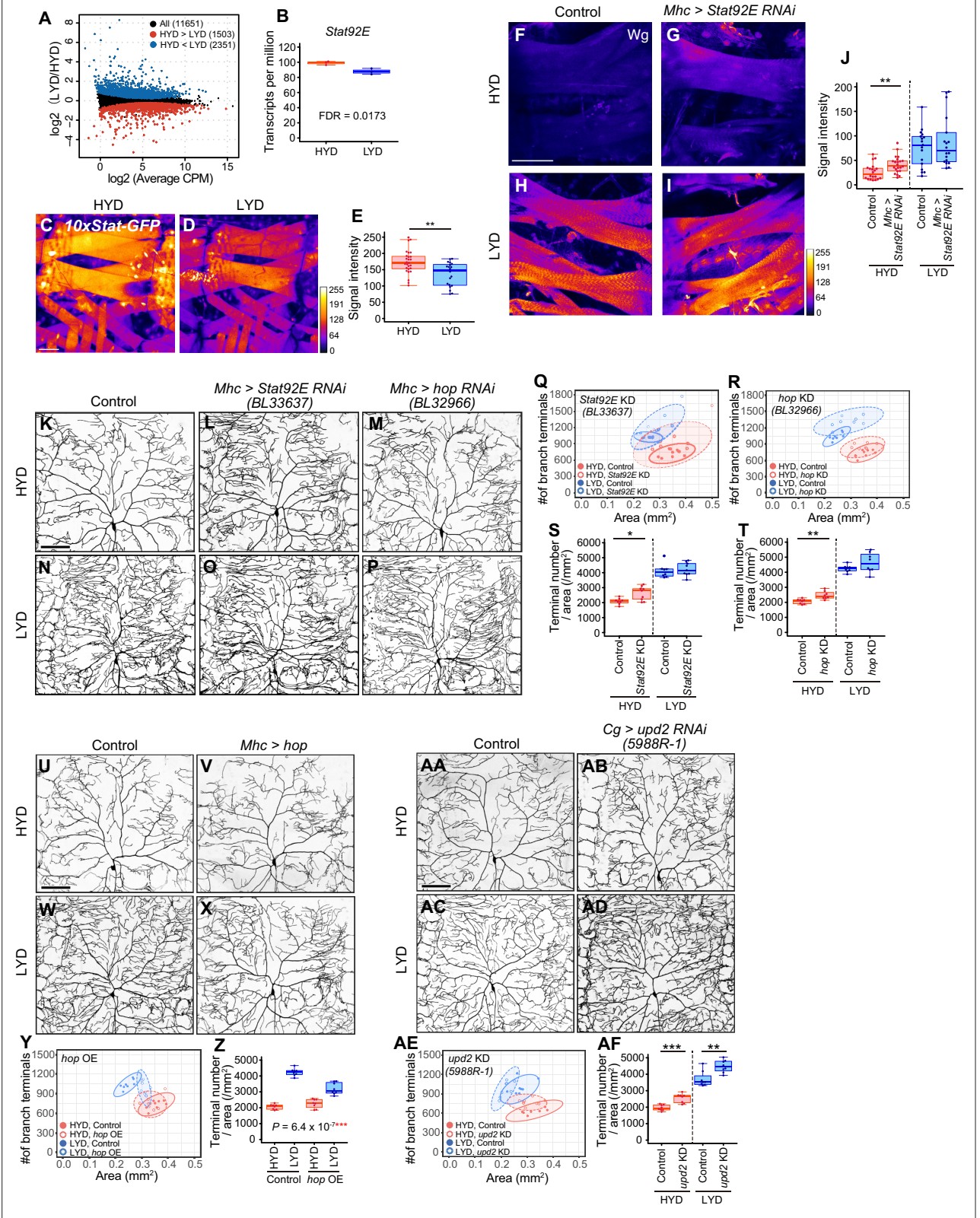

**Figure 5.** Downregulation of Wg expression by Stat92E on HYD suppresses the hyperarborization phenotype. (**A and B**) Plots of whole-body RNA-seq of wandering 3rd instar larvae reared on HYD or LYD. (**A**) The fold change (LYD/HYD) in read counts was plotted against average counts per million mapped reads (CPM) for HYD and LYD. Dots that are statistically supported (FDR ≤0.05) are colored (red for HYD >LYD and blue for HYD <LYD). (**B**) Plot of transcripts per million (TPM) of *Stat92E*. Adjusted p-value with Benjamini & Hochberg correction (FDR) is indicated. (**C–E**) Muscles of *10 x Stat-GFP*

*Figure 5 continued on next page*

*Figure 5 continued*

larvae on HYD (**C**) or LYD (**D**). The signal intensities of GFP correspond to the indicated color code. (**E**) Quantification of 10 x Stat-GFP intensity in muscle 9 (Student's t-test, n=20–27). (**F–J**) Muscles of control larvae (**F and H**) or larvae with *Stat92E* KD in muscles (**G and I**) on HYD or LYD were stained for Wg. The signal intensities correspond to the indicated color code. (**J**) Quantification of the mean Wg immunofluorescence in muscle 9 (Wilcoxon-Mann-Whitney test, n=17–22). (**K–T**) Images of ddaC neurons in control larvae (**K and N**), larvae with *Stat92E* KD in muscles (**L and O**), or larvae with *hop* KD in muscles (**M and P**), on HYD or LYD. 2D plots (**Q and R**) and densities of branch terminals (S and T, Wilcoxon-Mann-Whitney test, n=8–9). (**U–Z**) Images of ddaC neurons in control larvae (**U and W**) or larvae with *hop* overexpression in muscles (**V and X**), on HYD or LYD. 2D plot (**Y**) and densities of branch terminals (Z, two-way ANOVA, n=8). (AA-AF) Images of ddaC neurons in control larvae (AA and AC) or larvae with *upd2* KD in the fat body and hemocytes (AB and AD), raised on HYD or LYD. 2D plot (AE) and densities of branch terminals (AF, Wilcoxon-Mann-Whitney test, n=8). The ddaC neurons were visualized by expressing *ppk-CD4:tdGFP*. Control data in (**R**) and (**T**) are shared with (**Y**) and (**Z**). Boxplots in (**B, E, J, S, T, Z and AF**) are depicted as in *Figure 1C*. *p<0.05, **p<0.01, and ***p<0.001. Scale bars, 100 μm.

The online version of this article includes the following figure supplement(s) for figure 5:

**Figure supplement 1.** Enriched terms in functional annotation clustering of differentially expressed genes depending on diets in whole body RNA-seq data.

**Figure supplement 2.** Effects of inhibiting components of JAK/STAT pathway on hyperarborization.

HYD (*Figure 5F–J*; compare 5 F with 5 G). Furthermore, knocking down *Stat92E* or *hopscotch* (*hop*) encoding JAK in muscles promoted hyperarborization of C4da neurons in larvae reared on HYD (*Figure 5K–T* and *Figure 5—figure supplement 2A, B, F, G, K, O*). In contrast, overexpression of *hop* in muscles ameliorated the hyperarborization on LYD (*Figure 5U–Z*). These results indicate that JAK/STAT signaling contributes to the downregulation of *wg* expression in muscles and the suppression of dendritic hyperarborization on HYD (Figure 7).

It was previously reported that Upd2 secreted from the fat body activates JAK/STAT signaling through transmembrane receptor Domeless (Dome) in GABAergic neurons in the adult brain, which project onto insulin producing cells (IPCs), thereby regulating systemic growth in a nutritional-status-dependent manner (*Rajan and Perrimon, 2012*). It has also been shown that the secretion of Upd2 or Upd3 from hemocytes promotes the expression of a Stat92E reporter in larval muscle (*Yang et al., 2015*). These studies prompted us to address whether any Upds from the fat body or hemocytes, and Dome in muscles, contribute to the hyperarborization phenotype. Knocking down *upd2*, but not *upd* or *upd3*, in the fat body and hemocytes resulted in an increased terminal density on HYD (*Figure 5AA-AF* and *Figure 5—figure supplement 2S-AN*). This effect of *upd2* KD in the fat body and hemocytes is similar to that of *Stat92E* or *hop* KD (*Figure 5K–T*) and that of *wg* overexpression in muscles (*Figure 3T–Y*). Enhanced branching on HYD was also seen in a *dome* KD in one out of three RNAi lines (*Figure 5—figure supplement 2C–E, H-J, L-N and P-R* ). Although it is necessary to verify KD of *dome* in the future, these results are suggestive of the role of fat body (and hemocytes)–muscle inter-organ communication through a Upd2-Stat92E pathway in suppressing the hyperarborization phenotype on HYD. To address whether the key nutrients (vitamins, metal ions, and cholesterol) increase Stat92E expression in muscles, we examined the reporter expression on LYD supplemented with or without VMC. However, the addition of VMC to LYD did not increase the signal intensity of the Stat92E reporter (data not shown). This result contrasts with the decreased level of Wg in response to the key nutrients (*Figure 3M–P*). The *Stat92E* KD caused only a marginal Wg increase on HYD compared to the difference in the amount of Wg between HYD and LYD (*Figure 5F, H and J*). Considering these results, it is likely that an additional unknown molecular mechanism other than the JAK/STAT pathway contributes to the high VMC-mediated downregulation of Wg in muscles (Figure 7).

## LYD blunts light responsiveness of C4da neurons and larval light avoidance behavior

C4da neurons sense noxious thermal, mechanical, and light stimuli (*Chin and Tracey, 2017*). We therefore examined how our dietary conditions affect the electrophysiological activity of C4da neurons and larval behavior (*Figure 6*). First, we compared firing activities of C4da neurons in larvae that were reared on either HYD or LYD. As a noxious stimulus, we illuminated entire arbors of recorded neurons with blue light (*Xiang et al., 2010*; *Terada et al., 2016*). We used extracellular recording to monitor both spontaneous and evoked activities (*Figure 6A–C*). The frequency of spontaneous firing was higher in C4da neurons from larvae reared on LYD than on HYD (*Figure 6D*). Regarding the response to the light stimulus, all relevant parameters, i.e., the firing frequency, the change amount, and the

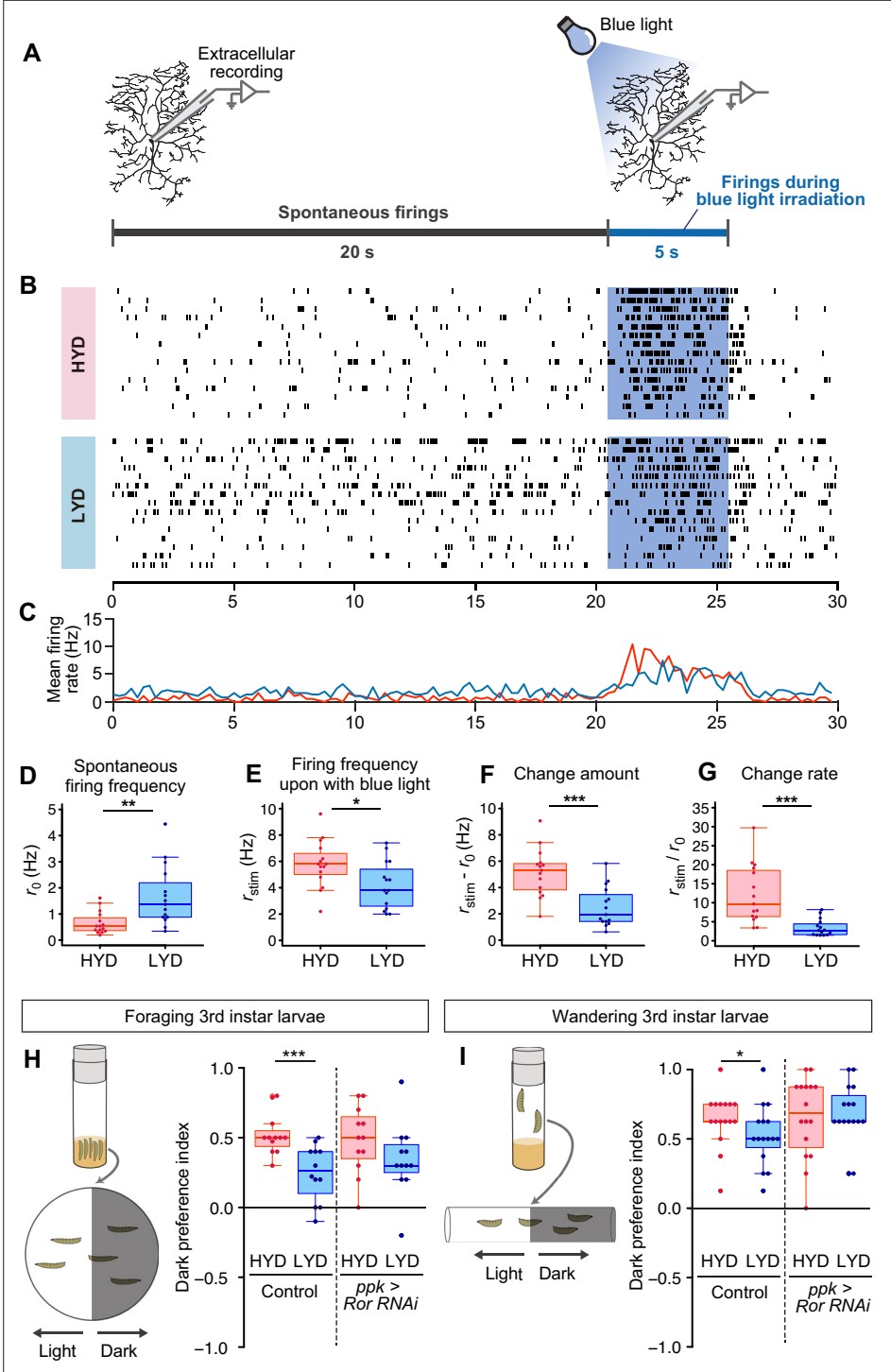

**Figure 6.** LYD blunts light responsiveness of C4da neurons and larval light avoidance behavior. (**A**) A schematic diagram outlining the electrophysiological analysis. Firing activities of C4da neurons v'ada were recorded by measuring the extracellular membrane potential. After spontaneous firings were recorded for about 20 s, activities during blue light irradiation were monitored for 5 s. (**B and C**) Firing activities of C4da neurons on HYD or LYD. (**B**) Raster plots of firing. Blue shading indicates the 5 s blue light irradiation. Each row in the plots represents the data for a single cell. (**C**) Peristimulus time histograms calculated at 250 ms bins, on HYD (red line) or LYD (blue line). (**D–G**) Quantitative analysis of the firing activities. (**D**) Spontaneous firing frequency. (**E**) Firing frequency during blue light irradiation. (**F**) Change amount of the firing response to the blue light stimulus calculated by subtracting [spontaneous firing frequency] from [firing frequency during blue light irradiation] (**G**) Change rate of

*Figure 6 continued on next page*

*Figure 6 continued*

the firing response to the blue light stimulus calculated by dividing [firing frequency during blue light irradiation] by [spontaneous firing frequency]. (Wilcoxon-Mann-Whitney test, n=15). (**H and I**) Schematic diagram of light/dark choice assays and dark preference index of foraging 3rd instar larvae on agar plates (**H**), and wandering 3rd instar larvae in plastic tubes (**I**). Control larvae and larvae with *Ror* KD in C4da neurons were tested. (Wilcoxon-Mann-Whitney test, n=12–16) Boxplots in (**D–I**) are depicted as in *Figure 1C*. *p<0.05, **p<0.01, and ***p<0.001.

change rate, were lower on LYD than on HYD (*Figure 6E–G*, see definition of the parameters in the legend), indicating that C4da neurons on LYD are less sensitive to the stimulus. Next, we examined whether the blunted light responsiveness of the neurons affects larval avoidance behavior. It was reported that *Drosophila* larvae prefer dark places to avoid noxious light, and this light avoidance behavior requires the activity of C4da neurons (*Yamanaka et al., 2013*; *Imambocus et al., 2022*). We speculated that the blunted light responsiveness of larvae on LYD may cause declines in their light avoidance behavior, and this may allow larvae to continue their search for high-nutrient food. To address this possibility, we conducted light/dark choice assays in which larvae reared on HYD or LYD were allowed to choose between dark and bright areas. We found that both foraging 3rd instar

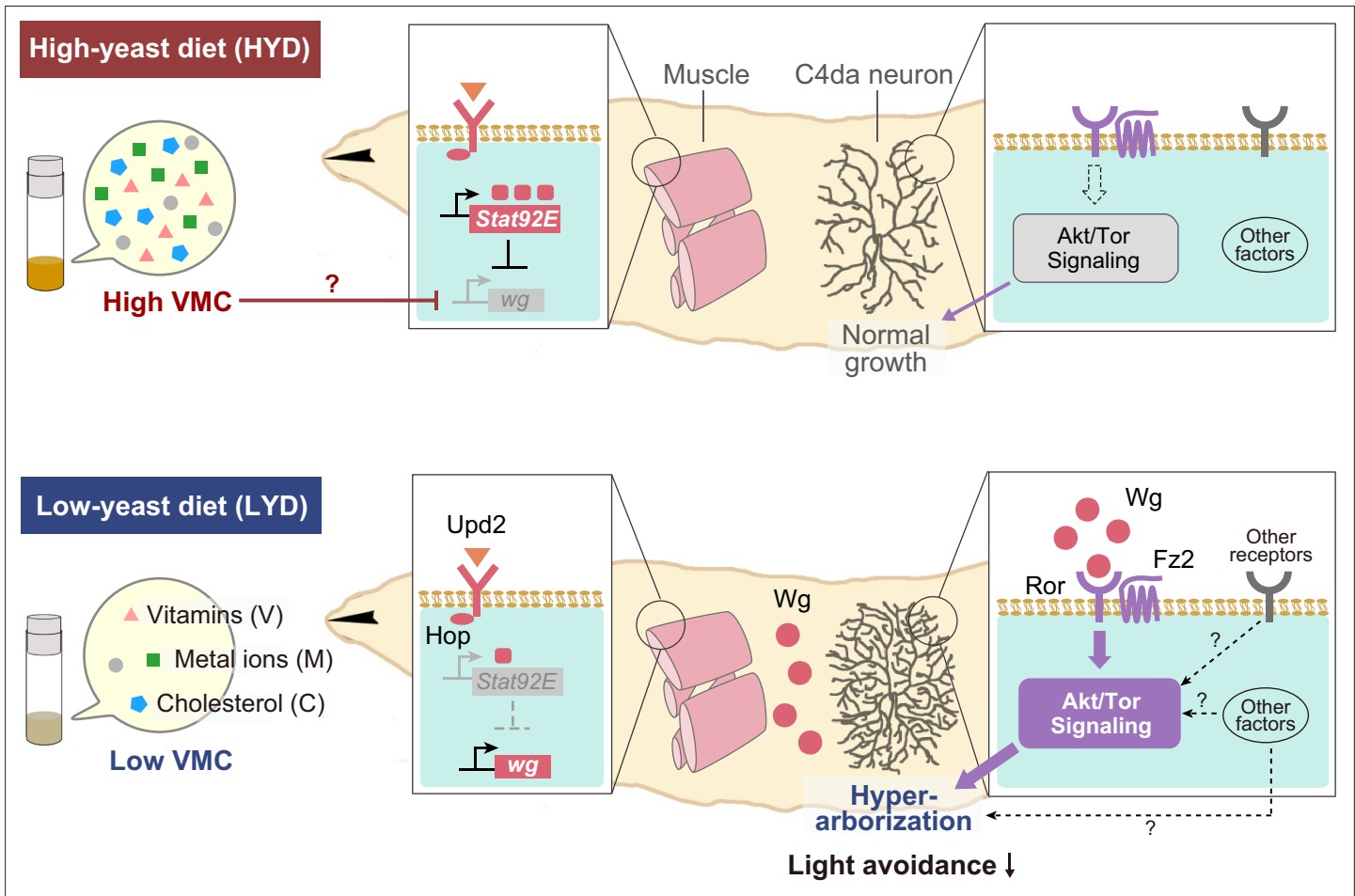

**Figure 7.** Model of the low-nutrient dependent dendritic hyperarborization. Compared to C4da neurons of HYD-fed larvae (top), those of LYD-fed larvae (bottom) increase in dendritic branching (hyperarborization) due to a combined deficiency of vitamins, metal ions, and cholesterol in the food ('Low VMC'). In the LYD-fed larvae, *wg* expression in muscles is higher than in muscles in the HYD-fed larvae; secreted Wg is bound by receptors Ror and Fz2 in C4da neurons, which in turn hyperactivate Akt signaling, thereby promoting dendrite branching. In the C4da neurons, other receptors (e.g. InR or Alk) upstream of Akt and intracellular components of Wnt signaling ('Other factors' such as Dsh and Bsk) might also contribute to the hyperarborization phenotype. The cellular response of C4da neurons is associated with a whole-animal level response (blunted light avoidance behavior). In the HYD-fed larvae, *wg* expression is suppressed partly by Upd2-Hop-Stat92E signaling and partly by an abundant VMC-mediated unknown molecular mechanism (red T bar and '?'). See Results and Discussion for details.

and wandering 3[rd] instar larvae on LYD showed lower preference for dark places than larvae on HYD (*Figure 6H and I*). Furthermore, when *Ror* was knocked down in C4da neurons, differences in light avoidance behavior between the diets tended to be smaller than the control larvae (*Figure 6H and I*). Our results suggest that the hyperarborization of C4da neurons is associated with blunted light avoidance behavior.

## Discussion

Collectively, our studies illustrate how selective nutrients in the food impact neuronal development through inter-organ signaling (*Figure 7*). Yeast has long been considered as a rich source of amino acids for *Drosophila*; however, our results suggest that C4da neurons increase their dendritic terminal density on the LYD due to a combined deficiency in vitamins, metal ions, and cholesterol. This result is unexpected, because previous studies on nutrition-dependent cell growth has focused primarily on the TOR signaling pathway, which is activated by amino acids (*González et al., 2020*; *Liu and Sabatini, 2020*). The addition of the above nutrient trio to LYD did not restore the branch terminal density to the same level as HYD. This may indicate that the balance of concentrations among these nutrients was not optimized or that unknown nutrients may need to be added along with these nutrients. In addition, we showed that up-regulation of Wg expression in muscle on LYD was suppressed by supplementation of these components to the diet. This regulation may be achieved at least at a transcriptional level possibly by an interplay between a hypothetical nutrient-responsive module in the cis-element of *wg* and transcription factors and/or epigenetic machineries that require vitamins, vitamin-derived metabolites, metal ions and cholesterol (*Harris et al., 2016*; *Kambe et al., 2016*). Further investigations are necessary to understand the detailed molecular mechanisms underlying the combined effects of these components on *wg* expression. It has been reported that increasing or decreasing the amount of yeast in foods causes various responses in *Drosophila* (*Bass et al., 2007*; *Okamoto and Nishimura, 2015*), and the approach used in this study may help to clarify which nutrients are the key factors that cause those responses.

### Inter-organ Wg/Ror/Akt signaling-mediated the hyperarborization phenotype

Previous studies demonstrated the coordinate growth control of dendrites of C4da neurons and the epidermis (*Parrish et al., 2009*; *Jiang et al., 2014*), from which a separate model underlying dendritic hyperarborization has evolved (*Poe et al., 2020*). In that model, stress sensor FoxO is expressed less in neurons than in neighboring epidermal cells, which results in lower levels of autophagy and less suppression of Tor signaling in the neuron, thereby ensuring dendritic growth even under a low-yeast condition (*Poe et al., 2020*). In contrast to the above model, our model highlights the signaling between neurons and another adjacent tissue: muscles secrete Wg, while C4da neurons express the receptor complex Ror-Fz2 on their cell surface. Therefore, it is likely that both the extrinsic Wg-dependent mechanism and the FoxO-dependent intrinsic sparing mechanism work together to generate the hyperarborization phenotype.

Ror is mainly expressed in the nervous system. No significant abnormalities in neuronal morphology including that of the C4da neurons were observed in *Ror* mutants under standard dietary conditions (*Ripp et al., 2018*; *Nye et al., 2020*). Consistent with these reports, there was no significant difference in morphological features of dendritic arbors of C4da neurons between *Ror* mutant or KD larvae and control larvae under the nutrient-rich HYD condition (*Figure 2L and R*). Because Ror is required for the hyperarborization under the hypotrophic condition (LYD in this study; *Figure 2N and T*) and for dendrite regeneration through the regulation of microtubule nucleation (*Nye et al., 2020*), it could be an adaptive agent that copes with environmental stress or damage. The pathway for microtubule nucleation includes Dsh and Axn, whose KD appeared to ameliorate the hyperarborization of C4da neurons. It remains to be seen how Dsh and Axn contribute to the hyperarborization on LYD, and whether the Wg-Ror-Akt pathway and the Ror-mediated microtubule-nucleation mechanism intersect. A number of other RTKs, such as insulin/IGF receptors and EGFR, activate Akt in other cellular contexts such as growth and proliferation of various stem cells and mammalian cancer cells (*Shim et al., 2013*; *Butti et al., 2018*). It is likely that InR also functions upstream of Akt in C4da neurons (*Parrish et al., 2009*; *Shimono et al., 2014*; *Poe et al., 2020*). Future studies will explore whether InR

and other RTKs indeed function upstream of Akt in the context of the Wg/Ror/Akt signaling, and if so, how these various inputs are integrated by Akt to realize the nutritional status-dependent dendrite branching of C4da neurons (*Figure 7*).

The muscle is not only an energy-consuming organ, but it also plays an important role in regulating metabolic signaling through inter-organ communication with other tissues such as the brain and the fat body (*Bretscher and O'Connor, 2020*). In the adult stage, for example, muscle-derived Wg regulates lipid storage in the fat body (*Lee et al., 2014*). Our study revealed that muscle-derived Wg, which is up-regulated in response to low levels of vitamins, metal ions, and cholesterol, regulates dendrite branching of C4da neurons in the larval stage. Therefore, the muscle functions as a mediator of the nutritional status to other peripheral tissues in both growing and adult stages, and Wnt signaling may play a pivotal role in fulfilling this metabolic response function throughout the life cycle.

In our search for regulatory mechanisms of *wg* expression, we found that Stat92E reporter expression was higher in muscles on HYD than on LYD. This finding is reminiscent of Stat92E reporter expression in a population of GABAergic neurons in the adult brain, which project onto the Insulin producing cells (IPCs) and inhibit the release of Dilps (*Rajan and Perrimon, 2012*). This reporter expression in the GABAergic neurons also varies in a nutritional-status-dependent manner: the expression is higher on a standard laboratory food containing yeast compared to a sucrose-only condition. Our results suggest that Upd2-Stat92E signaling partially contributes to downregulation of Wg in muscles and suppression of the hyperarborization phenotype. When *stat92E* was knocked down in muscles, Wg increased on HYD; however, that increase was marginal compared to the difference in the level of Wg expression between HYD and LYD (*Figure 5F, G, H and J*). Moreover, the addition of a combination of vitamins, metal ions, and cholesterol to LYD did not increase the signal intensity of a Stat92E reporter (data not shown). Therefore, it is likely that in the HYD-fed larvae, an unknown VMC-mediated molecular mechanism also contributes to downregulation of *wg* in muscles (red T bar with '?' in *Figure 7*). Further investigation is required to elucidate how Wg expression in muscles is controlled at the molecular level in such a key nutrient(s)-dependent manner and how the Upd2-Stat92E pathway contributes to the entire mechanism of inter-organ communication.

## Physiological roles of C4da neurons and the hyperarborization phenotype

What are the implications of the inter-organ signaling mechanism controlling dendritic branches in the context of nutritional adaptation? It has been reported that a wide range of animals tend to take more risks when they are hungry (*Symmonds et al., 2010*; *Filosa et al., 2016*; *Padilla et al., 2016*; *Bräcker et al., 2013*). Our electrophysiological analysis indicates that C4da neurons on LYD decrease light-evoked response (*Figure 6A–G*). Consistently, larvae reared on LYD displayed decreases in their dark preference, compared to those reared on HYD, and they explored bright places, which is potentially risky for their survival (Control of *Figure 6H and I*). This difference between the diets tended to become smaller once the inter-organ signaling mechanism was suppressed in C4da neurons (*ppk >Ror* RNAi in *Figure 6H and I*). These results imply that the hyperarborization of C4da neurons on LYD might contribute to blunting light avoidance behavior, although we cannot exclude the possibility that the *Ror* KD might affect neuronal functions through other mechanisms than the dendritic morphological change. Our study raises the possibility that nutrient-dependent development of somatosensory neurons plays a role in optimizing a trade-off between searching for high-nutrient foods and escaping from noxious environmental threats. Although a recent study described the circuitry required for the larval light avoidance behavior, it remains unclear whether the possible modifications of neural circuits downstream of C4da neurons take part in this behavioral transition (*Imambocus et al., 2022*). The identification of the downstream circuits would allow further study of the relationship between nutrient-dependent neural differentiations and evoking risk-taking behavior.

In contrast to our results, a previous study reported that larvae with hyperarborized C4da neurons react more quickly to noxious heat (*Poe et al., 2020*). While light-induced $Ca^{2+}$ activity in C4da neurons decreases, thermal nociceptive behavior increases during $2^{nd}$ and $3^{rd}$ instar larval periods (*Jaszczak et al., 2022*), which indicates that these nociceptive responses are regulated in the opposite direction or in distinct fashions. Therefore, seemingly contradictory results between the previous study and ours may be due to different regulatory mechanisms of the sensory modalities.

The relationship between nutritional status and neural development has often been studied epidemiologically (*Prado and Dewey, 2014*; *Bhutta et al., 2017*). Our study, which presents a mechanism by which quantitative changes in specific nutrients act on neuronal morphology and operate through inter-organ signaling, provides a stepping stone for future explorations of molecular mechanisms linking nutrition and development of other neuronal cell types and in other animal species.

## Materials and methods

### *Drosophila* strains and fly culture

Fly strains used in this study are listed in Key Resources Table. Our stocks are usually reared on a laboratory standard diet (*Watanabe et al., 2017*). Adult males and virgin females that had developed on the standard diet were collected and crossed on the standard diet for 3–5 days. Then, the adults were transferred into vials containing HYD or LYD, which were identical to the semidefined medium (SDM)-based diet (8% Y) and the SDM-based diet (0.8% Y), respectively (see *Supplementary file 1* and its legend in *Watanabe et al., 2017*). After an egg-laying interval, the adult flies were cleared in every experiment and wandering 3rd instar larvae that came out of individual diets were used. Larvae were reared under noncrowded conditions at 25 °C in all the experiments except the *wg* overexpression experiments at 29 °C. Our recombinant DNA experiments follow Kyoto University Regulations for Safety Management in Recombinant DNA Experiments under protocol # 210059.

### Experimental diets

We cooked the high yeast diet (HYD) or low yeast diet (LYD) based on semidefined media (SDM) as described previously (*Watanabe et al., 2017*). The original SDM recipe is described at the Bloomington *Drosophila* Stock Center https://bdsc.indiana.edu/information/recipes/germanfood.html. HYD and LYD were composed of brewer's yeast (MPBio 2903312), glucose (Wako 049–31165), sucrose (Wako 196–00015), peptone (Fluka 82303), and agar (Matsuki Kanten). The complete compositions of these diets can be found in *Supplementary file 1*. After the ingredients were mixed, water was added to a final volume of 200 ml, followed by autoclaving. Once the foods had cooled, 1.2 ml propionic acid (Nacalai Tesque, 29018–55) and 2 ml 10% butyl p-hydroxybenzoate (Nacalai Tesque, 06327–02) in 70% ethanol were added. The foods were then dispensed into vials and left overnight before use.

For the supplementation with essential amino acid solution, we used 50 x MEM EAA solution (Wako 132–15641). Each fraction of holidic medium (*Piper et al., 2014*; *Piper et al., 2017*) other than amino acids (vitamins, cholesterol, metal ion and other ingredients) was added to LYD at 10 times the concentration in holidic medium. Amino acids mixture from holidic medium was added at a 1 x or 3 x concentration. The complete compositions of diets used in nutrient supplementation experiments can be found in *Supplementary file 1*.

### Imaging and quantification for assessing dendritic morphology

Images of ddaC (C4da) or ddaF (C3da) neurons in A3–A5 segments were acquired in live whole-mount larvae as described (*Hattori et al., 2013*; *Parrish et al., 2009*; *Matsubara et al., 2011*). Protocols for single-cell labeling (MARCM) were as previously described (*Shimono et al., 2014*). For quantification of the number of dendritic branching terminals, we drew an outline of the dendritic field as a region of interest (ROI) by connecting the outermost dendritic terminals with the Adobe Photoshop path tool. Then, dendritic branching terminals inside the ROI of ddaC were automatically counted using DeTerm (*Kanaoka et al., 2019*). Concerning counting short spikes of ddaF, results obtained by DeTerm were corrected manually. In addition, the area size of the ROI was measured as the dendritic coverage size. Dendrite length of ddaC neurons was measured using Fiji/ImageJ as previously described (*Poe et al., 2017*). Briefly, images of dendrites were processed sequentially by Gaussian Blur, Auto Local Threshold, Particles4, and Skeletonize (2D/3D), and the length of the 1-pixel-width skeleton was measured inside the ROI. Some representative control images and control data are shared by multiple figures. See figure legends.

### Preparation of larvae with developmental delay

In the experiments in which larval growth was delayed by excess sucrose, we collected eggs as previously described (*Watanabe et al., 2019*) and placed them on either HYD, LYD, or HYD +sucrose in

vials. The complete composition of HYD + sucrose can be found in *Supplementary file 1*. In a *dilp8* overexpression experiment (*Colombani et al., 2012*), the adult flies were allowed to oviposit on HYD or LYD for 24 hr. The timing of the neuronal observations under each condition is indicated in *Figure 1—figure supplement 2*.

## RTK screening

We conducted two rounds of screening. In the primary screening, we intended to enhance KD efficacy and used *Gr28b.c-GAL4* and *ppk-GAL4* together. We acquired images of 3–8 knocked-down neurons for each gene on each diet, and then visually judged whether hyperarborization was blunted or not. We selected nine genes (*Ror*, *InR*, *Alk*, *htl*, *Egfr*, *Pvr*, *Ddr*, *dnt*, and *drl*) for the secondary screening, in which we used only *ppk-GAL4* because *Gr28b.c-GAL4* is expressed in a small subset of neurons in the central nervous system in addition to C4da neurons in the peripheral nervous system (*Xiang et al., 2010*). See *Supplementary file 2* for names of the 20 RTK genes and stock numbers of RNAi lines used.

## Immunostaining

Dissected wandering 3rd instar larvae were fixed in a 1:10 dilution of Formaldehyde Solution (Nacalai Tesque, 16222–65) in PBS plus 0.05% Triton X-100 for 30 min, then washed three times in PBS plus 0.1% Triton X-100 (PBST). After blocking in PBST plus 2% bovine serum albumin for 30 min, primary antibodies listed in Key Resources Table were added, then incubated overnight at 4 °C. After three successive washes, secondary antibodies were added, then incubated for 1 hr at room temperature. Finally, samples were mounted using FluorSave Reagent (Calbiochem). Most of the images were acquired with a Nikon C1 laser scanning confocal microscope coupled to a Nikon Eclipse E-800 microscope. The images in *Figure 2—figure supplement 5* were acquired with a ZEISS LSM 800 microscope.

## Quantification of signal intensity

To quantify signal levels in muscle, we made Z-stack images and chose muscle 9, one of the closest muscles to ddaC, for measuring the signal intensity. For quantification of Wg or 10 x Stat-GFP signals, we measured the signal intensity inside a 19 μm or a 27 μm square ROI, respectively. Three ROIs were drawn for each muscle, and the average value was calculated. For quantification of RedStinger driven by *wg-GAL4*, the signal intensity inside nuclei that were identified by DAPI signals was measured. Then, the values from 2 to 7 nuclei in each muscle were averaged. For quantification of p-Akt levels in cell bodies of C4da ddaC, C1da ddaE or C3da ddaF neurons, we selected the single section containing the strongest signal in the neurons and measured signal intensities inside a 1.7 μm square ROI located on the abdominal side of nuclei of the neurons. However, for the quantification in ddaF neurons, the ROI was placed on the ventral side of nuclei only when we could not identify the border between ddaF and ddaC.

## Electrophysiology

Extracellular single-unit recordings in wandering 3rd instar larvae were performed as previously described (*Terada et al., 2016*; *Onodera et al., 2017*). We recorded the activity of v'ada of C4da neurons, which showed hyperarborization on LYD. For blue light irradiation, the 460–495 nm light at 72mW/mm$^2$ power was projected onto larvae for 5 s. The light spot was 1.5 mm in diameter. Peristimulus time histograms were calculated at 250 ms bins. The mean spontaneous firing frequencies were quantified in the 20 s window preceding the light stimuli. The mean firing frequencies during the light stimulation were quantified in the 5 s entire window. The firing changes were calculated by subtracting the mean spontaneous firing rate from the light-evoked one (Change amount) or using a ratio of the mean spontaneous firing rate divided by the mean light-evoked one (Change rate).

## Light/dark choice assay

Light/dark choice assays were performed as previously described with modifications (*Yamanaka et al., 2013*). For the assay of foraging larvae, we used foraging 3rd instar larvae one day before they start wandering. We prepared 2% agar plate with a lid half of which was covered with black tape and 20 larvae were placed along the junction between light and dark sides. After the plates were illuminated

for 15 min with white LED light (OHM, ODS-LKL6-W) at 700 lux, the number of larvae in both dark and light areas were counted. In some trials, one or two larvae dug into the agar. Such larvae were excluded from calculation of dark preference index. For the assay of wandering larvae, two opposed plastic tubes were joined by transparent scotch tape and one of the vials was covered with black tape. After 16 wandering larvae reared on HYD or LYD were put near the junction of the tubes, they were illuminated by the 700 lux light for 15 min, then the number of larvae in both dark and light areas were counted. Dark preference index was calculated as follows:

$$\big((\text{Number of larvae in dark}) - (\text{Number of larvae in light})\big) / (\text{Total number of larvae})$$

## RNA-seq

Protocols for sample preparation and data analysis of RNA-seq were essentially as described in *Watanabe et al., 2019*. To prepare each replicate, RNA was extracted from five whole bodies of male wandering third-instar larvae. The following procedures are different from *Watanabe et al., 2019*: (1) the NEBNext Ultra II Directional RNA Library Prep Kit for Illumina (NEB, E7760) was used for library preparation. (2) RNA-sequencing was performed on an Illumina NextSeq 500 system using single end reads. (3) All raw sequencing data were trimmed using TrimGalore (ver. 0.6.0, Cutadapt ver. 1.18; DOI:10.5281/zenodo.5127899, DOI:10.14806/ej.17.1.200) with -clip_R1 13 option. (4) Gene-based read counts were obtained using htseq-count (ver. 0.11.3; *Anders et al., 2015*) with -s reverse -a 10 options. (5) Differential expression analysis was performed on the count data using a generalized linear model (GLM) in the edgeR Bioconductor package (ver. 3.30.3; *McCarthy et al., 2012*; *Robinson et al., 2010*). All the RNA-sequencing data have been deposited and are available in the DDBJ Sequence Read Archive. The accession numbers for the data are DRR311224-DRR311229 (BioProject accession number: PRJDB12048).

## Statistical analysis

R (R Core Team) was used for stastical analysis. Values of $P<0.05$ were considered statistically different. Student's t-test or the Wilcoxon-Mann-Whitney test was used for two-group comparisons, and Dunnett's test, Steel test, or Steel-Dwass tests were used for multiple comparisons. We used two-way analysis of variance (ANOVA) to analyze interactive effects between genotype and diet. On the other hand, we used the two-group comparison tests (Student's t-test or the Wilcoxon-Mann-Whitney test) when we simply focused on whether a genetic manipulation itself affected dendrite branching on the same diet. Statistical tests used, the exact sample size (n), and p values are shown in *Supplementary file 4*. R was also used to draw 95% confidence ellipses. See also figure legends for details.

## Acknowledgements

The reagents and genomic datasets were provided by the *Drosophila* Genetic Resource Center at Kyoto Institute of Technology, National Institute of Genetics, the Bloomington Stock Center, Vienna *Drosophila* Resource Center, FlyBase, and the Developmental Studies Hybridoma Bank maintained by the University of Iowa. We thank T Kondo and Y Sando for performing RNA-sequencing; J A Hejna for polishing the manuscript; T Kambe, R Niwa, T Jovanic, N Yamanaka, S Goulas, Y Shimada-Niwa, N Okamoto and other members of the Uemura laboratory for discussions and their technical assistance; M M Rolls, A Wodarz, T Igaki, T Ito, M Nakamura, M Yamazaki, M Sato, P Leopold, Y Sanaki and T Nishimura for kindly providing reagents; and Y Xiang for sharing unpublished results.

## Additional information

### Funding

| Funder | Grant reference number | Author |
| --- | --- | --- |
| Japan Agency for Medical Research and Development | JP18gm1110001 | Tadashi Uemura |

| Funder | Grant reference number | Author |
|---|---|---|
| Japan Society for the Promotion of Science | 15H02400 | Tadashi Uemura |
| Japan Society for the Promotion of Science | 21H00251 | Yukako Hattori |
| Japan Society for the Promotion of Science | 21K06186 | Yukako Hattori |
| Japan Society for the Promotion of Science | 20J15084 | Yasutetsu Kanaoka |
| Japan Science and Technology Agency | JPMJFR2051 | Yukako Hattori |
| Naito Foundation | | Yukako Hattori |
| Japan Foundation for Applied Enzymology | | Yukako Hattori |

The funders had no role in study design, data collection and interpretation, or the decision to submit the work for publication.

## Author contributions

Yasutetsu Kanaoka, Resources, Data curation, Formal analysis, Funding acquisition, Investigation, Visualization, Methodology, Writing – original draft; Koun Onodera, Data curation, Formal analysis, Investigation, Visualization, Methodology, Writing – review and editing; Kaori Watanabe, Investigation, Methodology, Writing – review and editing; Yusaku Hayashi, Investigation; Tadao Usui, Methodology, Writing – review and editing; Tadashi Uemura, Conceptualization, Resources, Supervision, Funding acquisition, Methodology, Writing – original draft, Project administration, Writing – review and editing; Yukako Hattori, Conceptualization, Data curation, Formal analysis, Supervision, Funding acquisition, Investigation, Visualization, Methodology, Writing – original draft, Project administration, Writing – review and editing

## Author ORCIDs

Yasutetsu Kanaoka http://orcid.org/0000-0002-1835-3248
Koun Onodera http://orcid.org/0000-0002-4203-9865
Kaori Watanabe http://orcid.org/0000-0003-2887-3690
Tadao Usui http://orcid.org/0000-0002-0507-1495
Tadashi Uemura http://orcid.org/0000-0001-7204-3606
Yukako Hattori http://orcid.org/0000-0001-5977-8501

## Ethics

Our recombinant DNA experiments follow Kyoto University Regulations for Safety Management in Recombinant DNA Experiments under protocol # 210059.

## Decision letter and Author response

Decision letter https://doi.org/10.7554/eLife.79461.sa1
Author response https://doi.org/10.7554/eLife.79461.sa2

## Additional files

### Supplementary files

- Supplementary file 1. Compositions of the experimental diets.

- Supplementary file 2. Summary of the RTK knockdown screening.

- Supplementary file 3. RNA-seq data of larval whole bodies at the wandering third-instar stage on HYD or LYD. (A) List of Differentially expressed genes between HYD and LYD in whole larval bodies at the wandering third-instar stage (adjusted P value < 0.05). (B and C) List of functional annotation clusters that were significantly enriched (enrichment score ≥ 1.3) in genes highly expressed on HYD rather than on LYD (B) or genes highly 1385 expressed on LYD rather than on HYD (C).

- Supplementary file 4. Statistical details of experiments and a list of genotypes.

- MDAR checklist

## Data availability

All the RNA-sequencing data have been deposited and are available in the DDBJ Sequence Read Archive. The accession numbers for the data are DRR311224-DRR311229 (BioProject accession number: PRJDB12048).

The following dataset was generated:

| Author(s) | Year | Dataset title | Dataset URL | Database and Identifier |
|-----------|------|---------------|-------------|-------------------------|
| Hattori Y, Kanaoka Y, Uemura T | 2021 | Transcriptome analysis of male *Drosophila* larvae reared on two different diets | https://ddbj.nig.ac.jp/resource/sra-submission/DRA012492 | DDBJ Sequence Read Archive, DRR311224-DRR311229 |

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

# Appendix 1

## Appendix 1—key resources table

| Reagent type (species) or resource | Designation | Source or reference | Identifiers | Additional information |
|---|---|---|---|---|
| genetic reagent (*Drosophila melanogaster*) | ppk-GAL4 UAS-mCD8:GFP | **Grueber et al., 2007** (https://doi.org/10.1242/dev.02666) | N/A | |
| genetic reagent (*D. melanogaster*) | Gr28b.c-GAL4 UAS-mCD8:GFP | **Xiang et al., 2010** (https://doi.org/10.1038/nature09576) | N/A | |
| genetic reagent (*D. melanogaster*) | ppk-CD4-tdGFP[1b] | **Han et al., 2011** (https://doi.org/10.1073/pnas.1106386108) | N/A | |
| genetic reagent (*D. melanogaster*) | ppk-CD4-tdGFP[8] | **Han et al., 2011** (https://doi.org/10.1073/pnas.1106386108) | N/A | |
| genetic reagent (*D. melanogaster*) | ppk-CD4-tdTom[4a] | **Han et al., 2011** (https://doi.org/10.1073/pnas.1106386108) | N/A | |
| genetic reagent (*D. melanogaster*) | Ror[4] | **Ripp et al., 2018** (https://doi.org/10.1242/bio.033001) | N/A | |
| genetic reagent (*D. melanogaster*) | ppk-GAL4 | Bloomington *Drosophila* Stock Center | Stock #: 32079 | |
| genetic reagent (*D. melanogaster*) | Mhc-GAL4 | **Schuster and Davis, 1996** (https://doi.org/10.1016/s0896-6273(00)80,197x) | N/A | |
| genetic reagent (*D. melanogaster*) | Cg-GAL4 | Bloomington *Drosophila* Stock Center | Stock #: 7011 | |
| genetic reagent (*D. melanogaster*) | ap[ts78j] wg[1] | KYOTO Stock Center | Stock #: 107069 | |
| genetic reagent (*D. melanogaster*) | wg[l-8] cn[1] bw[1] speck[1] | KYOTO Stock Center | Stock #: 107019 | |
| genetic reagent (*D. melanogaster*) | wg-GAL4 | Bloomington *Drosophila* Stock Center | Stock #: 83627 | |
| genetic reagent (*D. melanogaster*) | UAS-RedStinger | Bloomington *Drosophila* Stock Center | Stock #: 8547 | |
| genetic reagent (*D. melanogaster*) | UAS-Akt RNAi (BL33615) | Bloomington *Drosophila* Stock Center | Stock #: 33615 | |
| genetic reagent (*D. melanogaster*) | UAS-Akt RNAi (v2902) | Vienna *Drosophila* Resource Center | Stock #: 2902 | |
| genetic reagent (*D. melanogaster*) | UAS-Ror RNAi | National Institute of Genetics | Stock #: 4926 R-1 | |
| genetic reagent (*D. melanogaster*) | UAS-wg RNAi | Vienna *Drosophila* Resource Center | Stock #: 6692 | |
| genetic reagent (*D. melanogaster*) | UAS-Stat92E RNAi (BL33637) | Bloomington *Drosophila* Stock Center | Stock #: 33637 | |
| genetic reagent (*D. melanogaster*) | UAS-hop RNAi | Bloomington *Drosophila* Stock Center | Stock #: 32966 | |
| genetic reagent (*D. melanogaster*) | UAS-upd2 RNAi (5988 R-1) | National Institute of Genetics | Stock #: 5988 R-1 | |
| genetic reagent (*D. melanogaster*) | UAS-hop | Bloomington *Drosophila* Stock Center | Stock #: 79033 | |
| genetic reagent (*D. melanogaster*) | UAS-myrAkt | Bloomington *Drosophila* Stock Center | Stock #: 50758 | |
| genetic reagent (*D. melanogaster*) | UAS-wg.H.T:HA1 | KYOTO Stock Center | Stock #: 108488 | |
| genetic reagent (*D. melanogaster*) | 10XSTAT92E-GFP | Bloomington *Drosophila* Stock Center | Stock #: 26197 | |
| genetic reagent (*D. melanogaster*) | UAS-InR RNAi (BL31594) | Bloomington *Drosophila* Stock Center | Stock #: 31594 | |
| genetic reagent (*D. melanogaster*) | UAS-InR RNAi (BL51518) | Bloomington *Drosophila* Stock Center | Stock #: 51518 | |

*Appendix 1 Continued on next page*

*Appendix 1 Continued*

| Reagent type (species) or resource | Designation | Source or reference | Identifiers | Additional information |
|---|---|---|---|---|
| genetic reagent (*D. melanogaster*) | UAS-grnd RNAi | Vienna *Drosophila* Resource Center | Stock #: 43454 | |
| genetic reagent (*D. melanogaster*) | UAS-wgn RNAi | Vienna *Drosophila* Resource Center | Stock #: 9152 | |
| genetic reagent (*D. melanogaster*) | UAS-mth RNAi | Vienna *Drosophila* Resource Center | Stock #: 102303 | |
| genetic reagent (*D. melanogaster*) | UAS-babo RNAi | Bloomington *Drosophila* Stock Center | Stock #: 25933 | |
| genetic reagent (*D. melanogaster*) | UAS-Tor.TED | Bloomington *Drosophila* Stock Center | Stock #: 7013 | |
| genetic reagent (*D. melanogaster*) | UAS-S6k.KQ | Bloomington *Drosophila* Stock Center | Stock #: 6911 | |
| genetic reagent (*D. melanogaster*) | UAS-Tif-IA RNAi | Vienna *Drosophila* Resource Center | Stock #: 20336 | |
| genetic reagent (*D. melanogaster*) | UAS-Thor RNAi | Vienna *Drosophila* Resource Center | Stock #: 35439 | |
| genetic reagent (*D. melanogaster*) | UAS-foxo RNAi | Bloomington *Drosophila* Stock Center | Stock #: 32427 | |
| genetic reagent (*D. melanogaster*) | UAS-Alk RNAi (v11446) | Vienna *Drosophila* Resource Center | Stock #: 11446 | |
| genetic reagent (*D. melanogaster*) | UAS-Alk RNAi (BL107083) | Vienna *Drosophila* Resource Center | Stock #: 107083 | |
| genetic reagent (*D. melanogaster*) | mef2-GAL4 | Bloomington *Drosophila* Stock Center | Stock #: 27390 | |
| genetic reagent (*D. melanogaster*) | R38F11-GAL4 | Bloomington *Drosophila* Stock Center | Stock #: 50014 | |
| genetic reagent (*D. melanogaster*) | UAS-fz RNAi | Vienna *Drosophila* Resource Center | Stock #: 43075 | |
| genetic reagent (*D. melanogaster*) | UAS-fz2 RNAi | National Institute of Genetics | Stock #: 9739 R-1 | |
| genetic reagent (*D. melanogaster*) | "GAL4$^{5-40}$ UAS-Venus:pm SOP-FLP#42; tubPGal80 FRT2A" | KYOTO Stock Center | Stock #: 109950 | |
| genetic reagent (*D. melanogaster*) | "w*; FRT2A" | KYOTO Stock Center | Stock #: 106623 | |
| genetic reagent (*D. melanogaster*) | "y w hs-flp; fz2$^{C2}$ FRT2A" | **Chen and Struhl, 1999** (https://doi.org/10.1242/dev.126.23.5441) | N/A | |
| genetic reagent (*D. melanogaster*) | UAS-dsh RNAi | Vienna *Drosophila* Resource Center | Stock #: 101525 | |
| genetic reagent (*D. melanogaster*) | UAS-bsk.DN | Bloomington *Drosophila* Stock Center | Stock #: 6409 | |
| genetic reagent (*D. melanogaster*) | UAS-DAAM RNAi | Vienna *Drosophila* Resource Center | Stock #: 24885 | |
| genetic reagent (*D. melanogaster*) | UAS-arm RNAi | Vienna *Drosophila* Resource Center | Stock #: 7767 | |
| genetic reagent (*D. melanogaster*) | UAS-norpA RNAi | Bloomington *Drosophila* Stock Center | Stock #: 31113 | |
| genetic reagent (*D. melanogaster*) | UAS-Axn RNAi | Bloomington *Drosophila* Stock Center | Stock #: 31703 | |
| genetic reagent (*D. melanogaster*) | UAS-Stat92E RNAi (BL31318) | Bloomington *Drosophila* Stock Center | Stock #: 31318 | |
| genetic reagent (*D. melanogaster*) | UAS-dome RNAi (v106071) | Vienna *Drosophila* Resource Center | Stock #: 106071 | |
| genetic reagent (*D. melanogaster*) | UAS-dome RNAi (BL32860) | Bloomington *Drosophila* Stock Center | Stock #: 32860 | |

*Appendix 1 Continued on next page*

*Appendix 1 Continued*

| Reagent type (species) or resource | Designation | Source or reference | Identifiers | Additional information |
|---|---|---|---|---|
| genetic reagent (*D. melanogaster*) | *UAS-dome RNAi* (BL34618) | Bloomington *Drosophila* Stock Center | Stock #: 34618 | |
| genetic reagent (*D. melanogaster*) | *UAS-upd RNAi* (BL33680) | Bloomington *Drosophila* Stock Center | Stock #: 33680 | |
| genetic reagent (*D. melanogaster*) | *UAS-upd RNAi* (v3282) | Vienna *Drosophila* Resource Center | Stock #: 3282 | |
| genetic reagent (*D. melanogaster*) | *UAS-upd2 RNAi* (BL33949) | Bloomington *Drosophila* Stock Center | Stock #: 33949 | |
| genetic reagent (*D. melanogaster*) | *UAS-upd3 RNAi* (BL32859) | Bloomington *Drosophila* Stock Center | Stock #: 32859 | |
| genetic reagent (*D. melanogaster*) | *UAS-upd3 RNAi* (BL28575) | Bloomington *Drosophila* Stock Center | Stock #: 28575 | |
| genetic reagent (*D. melanogaster*) | *rn-GAL4* | **St Pierre et al., 2002** (https://doi.org/10.1242/dev.129.5.1273) | N/A | |
| genetic reagent (*D. melanogaster*) | *UAS-dilp8* | **Colombani et al., 2012** (https://doi.org/10.1126/science.1216689) | N/A | |
| genetic reagent (*D. melanogaster*) | *UAS-htl RNAi* (v6692) | Vienna *Drosophila* Resource Center | Stock #: 6692 | |
| genetic reagent (*D. melanogaster*) | *UAS-htl RNAi* (BL35024) | Bloomington *Drosophila* Stock Center | Stock #: 35024 | |
| genetic reagent (*D. melanogaster*) | *UAS-Egfr RNAi* | Vienna *Drosophila* Resource Center | Stock #: 43267 | |
| genetic reagent (*D. melanogaster*) | *UAS-Pvr RNAi* | Vienna *Drosophila* Resource Center | Stock #: 13502 | |
| genetic reagent (*D. melanogaster*) | *UAS-Ddr RNAi* | Vienna *Drosophila* Resource Center | Stock #: 29720 | |
| genetic reagent (*D. melanogaster*) | *UAS-dnt RNAi* | National Institute of Genetics | Stock #: 17,559 R-3 | |
| genetic reagent (*D. melanogaster*) | *UAS-drl RNAi* | Bloomington *Drosophila* Stock Center | Stock #: 29602 | |
| genetic reagent (*D. melanogaster*) | *UAS-Eph RNAi* | Bloomington *Drosophila* Stock Center | Stock #: 28511 | |
| genetic reagent (*D. melanogaster*) | *UAS-otk RNAi* | Bloomington *Drosophila* Stock Center | Stock #: 25790 | |
| genetic reagent (*D. melanogaster*) | *UAS-sev RNAi* | Bloomington *Drosophila* Stock Center | Stock #: 31274 | |
| genetic reagent (*D. melanogaster*) | *UAS-btl RNAi* | Vienna *Drosophila* Resource Center | Stock #: 110277 | |
| genetic reagent (*D. melanogaster*) | *UAS-Cad96Ca RNAi* | Vienna *Drosophila* Resource Center | Stock #: 1089 | |
| genetic reagent (*D. melanogaster*) | *UAS-CG10702 RNAi* | Vienna *Drosophila* Resource Center | Stock #: 27052 | |
| genetic reagent (*D. melanogaster*) | *UAS-Drl-2 RNAi* | Vienna *Drosophila* Resource Center | Stock #: 40484 | |
| genetic reagent (*D. melanogaster*) | *UAS-Nrk RNAi* | Vienna *Drosophila* Resource Center | Stock #: 9653 | |
| genetic reagent (*D. melanogaster*) | *UAS-Ret RNAi* | Vienna *Drosophila* Resource Center | Stock #: 107648 | |
| genetic reagent (*D. melanogaster*) | *UAS-tor RNAi* | Vienna *Drosophila* Resource Center | Stock #: 36280 | |
| genetic reagent (*D. melanogaster*) | *UAS-Tie RNAi* | Vienna *Drosophila* Resource Center | Stock #: 26879 | |
| genetic reagent (*D. melanogaster*) | *Gal4*[19-12] | **Xiang et al., 2010** (https://doi.org/10.1038/nature09576) | N/A | |
| antibody | anti-Wg (Mouse monoclonal) | Developmental Studies Hybridoma Bank | Cat# 4D4, RRID: AB_528512 | IF (1:15) |

*Appendix 1 Continued on next page*

Appendix 1 Continued

| Reagent type (species) or resource | Designation | Source or reference | Identifiers | Additional information |
|---|---|---|---|---|
| antibody | anti-Futch (Mouse monoclonal) | Developmental Studies Hybridoma Bank | Cat# 22C10, RRID: AB_528403 | IF (1:20) |
| antibody | anti-phospho-Akt (Rabbit polyclonal) | Cell Signaling | Cat# 9271 S | IF (1:100) |
| antibody | anti-DsRed (Rabbit polyclonal) | Clontech | Cat# 632496 | IF (1:250) |
| antibody | anti-Mouse IgG Alexa Fluor 488 (Goat polyclonal) | Invitrogen | Cat# A11029 | IF (1:1000) |
| antibody | anti-Rabbit IgG Alexa Fluor 488 (Goat polyclonal) | Invitrogen | Cat# A11034 | IF (1:1000) |
| antibody | anti-Mouse IgG Alexa Fluor 546 (Goat polyclonal) | Invitrogen | Cat# A11030 | IF (1:1000) |
| chemical compound, drug | Brewer's Yeast | MPBio | Cat# 2903312 | |
| chemical compound, drug | Yeast extract | Sigma-Aldrich Fluka | Cat# 70161 | |
| chemical compound, drug | Peptone from casein, enzymatic digest | Sigma-Aldrich Fluka | Cat# 82303 | |
| chemical compound, drug | Glucose | Wako | Cat# 049–31165 | |
| chemical compound, drug | Sucrose | Wako | Cat# 196–00015 | |
| chemical compound, drug | MgSO$_4$ | Wako | Cat# 132–00435 | |
| chemical compound, drug | CaCl$_2$ | Wako | Cat# 031–00435 | |
| chemical compound, drug | agar | Matsuki Kanten | N/A | |
| chemical compound, drug | Agar Purified, powder | Nacalai Tesque | Cat# 01162–15 | |
| chemical compound, drug | propionic acid | Nacalai Tesque | Cat# 29018–55 | |
| chemical compound, drug | butyl p-hydroxybenzoate | Nacalai Tesque | Cat# 06327–02 | |
| chemical compound, drug | 50 x MEM Essential Amino Acids Solution | Wako | Cat# 132–15641 | |
| chemical compound, drug | BSA | Nacalai Tesque | Cat# 01863–77 | |
| chemical compound, drug | Alexa Fluor 488 Phalloidin | Invitrogen | Cat# A12379 | |
| chemical compound, drug | FluorSave Reagent | Calbiochem | Cat# 345789 | |
| chemical compound, drug | Formalin | Nacalai Tesque | Cat# 16222–65 | |
| chemical compound, drug | DAPI | Nacalai Tesque | Cat# 19178–91 | |
| software, algorithm | DeTerm | *Kanaoka et al., 2019* ( https://doi.org/10.1111/gtc.12700) | N/A | |
| software, algorithm | R | R Core Team | RRID:SCR_001905 | |
| software, algorithm | Fiji | NIH | RRID:SCR_002285 | |
| software, algorithm | Photoshop | Adobe | RRID:SCR_014199 | |
| other | LED desk light | OHM | Cat# ODS-LKL6-W | This LED light was used in Light/dark choice assay |

