## [Editor Report]

Nutrition profoundly affects neural development. The Uemura lab previously reported that C4da neurons elaborate complex dendrites when larvae grow on low-yeast diets, a phenomenon called neural sparing. In the current study, they define the molecular mechanism underlying the nutrition-mediated phenomenon and identify that the inter-organ Wingless/Ror/Akt pathway between the neuron and its adjacent muscles is necessary and sufficient to mediate dendrite over-branching in the low-yeast condition.

---

## [Decision Letter]

**Decision letter after peer review:**

Thank you for submitting your article "Inter-organ Wingless/Ror/Akt signaling regulates nutrient-dependent hype-arborization of somatosensory neurons" for consideration by *eLife*. Your article has been reviewed by 3 peer reviewers, and the evaluation has been overseen by a Reviewing Editor and K VijayRaghavan as the Senior Editor. The following individuals involved in the review of your submission have agreed to reveal their identities: Fengwei Yu (Reviewer #1); Cheng-Ting Chien (Reviewer #2).

Essential revisions:

One of the reviewers has consolidated and summarized our consultations on the individual reviews. These are given below. Please attend to each concern while responding. Please all see the individual reviews below.

1) Are Wg levels increased in muscles when Upd2, Dome, or stat93E KD is knocked down. Since wg transcription could be regulated in a complex way, the Dome/JAK/STAT pathway is likely insufficient to be the main or only regulator. This is also echoed by a result described in the Discussion that Stat92E was not elevated by VMC supplement in LYD. The authors should limit their description on the significance of these results such as the description in Abstract (and Figure 7) and conclusion on lines 276-279 in Results. Also, they could include more discussion on the possibility of other pathways.

2) Could authors compare the KD efficiency in two Akt RNAi lines (v2902 and BL33615) (or are they known in previous literature)? They also need to test the specificity of anti-Akt antibodies using these akt RNAi lines. Are anti-Akt signals gone when Akt is knocked down?

3) Was the dendrite hyper-arborization phenotype also observed in other types of da neurons? Did you see similar pAkt upregulation in other da neurons (class I-III) under the condition of LYD diets? If the authors have the results, it should be very nice to demonstrate the general and significant effects of their findings and mechanism.

4) The Akt mutant phenotype is quite different from the Ror mutant phenotype based on the images shown in Figure 2. It seems to affect more parameters of dendrite structure than Ror. The authors should elaborate on the phenotypic difference between Akt and Ror mutants, perhaps using different quantification approaches. It is therefore appears unlikely that Akt, which has a stronger and different phenotype from Ror, only mediates Ror signaling. Indeed, as the authors mentioned in the Discussion, Akt has previously been shown to regulate ddaC dendrite branching through signals from epithelial cells, and it has also been previously demonstrated, that, unlike Ror, Akt affects dendrite growth on standard diets (Parrish, 2009). The only data that suggests that Ror functions upstream of Akt are the pAkt staining shown in Figure 4. The authors need to confirm the specificity of the Akt antibody. In addition, the authors should address this concern by modifying the model in the Discussion (such as to include the possibility of excess growth by Ror knockdown neurons in LYD seems likely to be related to the function of Ror in mediating microtubule nucleation, the possibility of integrating InR and AlK in the pathway, which they have shown in this study in the supplemental data, as well as in a previous study (Poe, 2020)).

5) It looks like the major change for wg RNAi in muscle is a change in arbor area in LYD- indeed this often seems to drive or at least contribute to the phenotype (except in Akt loss). Is this because animals are overall larger? Since the arbors cover the body- is this increase in size related to overall size? If so, is the dendrite difference secondary to a much broader change in animal size? Are the same conclusions reached if a total number of terminals is shown?

6) The issue of lengthened developmental time on LYD is problematic and not sufficiently addressed. In a previous study on this topic (Poe), the continued growth of ddaC neurons was interpreted as resilience to malnutrition and shown to be controlled by lower levels of a stress transcription factor, foxo, than in other cells. In this interpretation, the ddaC neurons just continue to grow at their normal pace while the overall development of the animal is slowed. How is this view reconciled with the current model? Which conditions change developmental timing in this study? Do any of the genetic manipulations rescue developmental speed- and if so, is dendrite architecture changing secondarily to this?

7) In figure 6 – it would be helpful, if feasible speedily, to include Ror k/d in the electrophysiological assays. The behavioral assays look somewhat inconclusive in terms of whether Ror rescues the LYD phenotype.

8) On page 9 the authors mentioned screening 20 RTKs. It would be helpful to include this data. Certainly, two others they include – InR and Alk – look like they have similar phenotypes to Ror and it would be helpful to know the specificity.

9) There are two ways of statistics when manipulating gene activities (RNAi or OE) in HYD and LYD. Direct comparisons of two samples were used mostly in the main Figures (except Figure 5Z), while "difference in two differences" were shown mainly for several Supplemental figures (2, 3, 4, 5). It is not clear why the authors used two different ways. Are there specific reasons for doing so? Could not the direct comparisons of two samples (such as for main figures) could cope with descriptions in the text and easier to comprehend.

*Reviewer #1 (Recommendations for the authors):*

Nutrition profoundly affects neural development. The Uemura lab previously reported that C4da neurons elaborate complex dendrites when larvae grow on low-yeast diets, a phenomenon called neural sparing. In this current study, they elegantly show that vitamins, metal ions, and cholesterol, but not amino acids, are critical for these hyperaborization defects. They then define the molecular mechanism underlying the nutrition-mediated phenomenon. They identify that the Wingless/Ror/Akt pathway between the neuron and its adjacent muscles is necessary and sufficient to mediate dendrite over branching in the low-yeast condition. Moreover, they also identify a systematic Upd2-Stat92E pathway which is activated in muscles by the fat body. As a result, the low-yeast condition can unsensitized C4da neuron functions which may help larva to survive better under limited nutrition conditions. Overall, this is a comprehensive and important study that provides mechanistic insights into the question of how neurons over-branch in low-yeast diets. The experiments were well designed, and the results were properly interpreted. Most of the figures are well presented. Overall, this is an interesting and rigorously conducted study and the current version is ready to be published.

*Reviewer #2 (Recommendations for the authors):*

1. Akt knockdowns in two lines (v2902 and BL33615) display qualitative suppression of dendrite hyperarborization in Figure 2. In v2902 KD, while the dendritic arbor looks more normal in both HYD and LYD, the BL33615 KD showed quite a dramatic reduction even in HYD, and also in LYD, rendering both sets of data groups clustering on a distinct area (Figure 2I), instead of in between two controls (HYD and LYD) such as in Figure 2G. While these two lines have been used before, I am wondering if the BL33615 has a stronger KD effect and if the basal Akt is necessary for a fundamental process in normal dendrite development (which is still preserved in the v2902 line), irrespective of nutrient condition. Could authors compare the KD efficiency in these two lines (or are they known in previous literature)?

2. The wg signal derived from muscles is required for dendrite hyperarborization (Figure 3) is a very nice set of data, although the effect by overexpression of wg in muscle is limited in HYD (Figure 3Y). It could be due to further control such as from epidermal cells shown in Poe et al. (2020) *eLife*. The experiment was performed at 29C to increase Wg expression, and the dendrite morphology was not normal. I am not asking for more experiments, but the authors could describe/explain what is the results at 25C instead of using 29C for readers not familiar with the approach? Have the authors tried using 2 copies of UAS-wg to increase wg expression at 25C?

3. There are two ways of statistics used when manipulating gene activities (RNAi or OE) in HYD and LYD. Direct comparisons of two samples were used mostly in the main Figures (except Figure 5Z), while "difference in two differences" were shown mainly for several Supplemental figures (2, 3, 4, 5). It is not clear why the authors used two different ways. Are there specific reasons for doing so? I thought the direct comparisons of two samples (such as for main figures) could cope with descriptions in the text and be easier to comprehend.

4. The RNAi KD effects of Dome/JAK/STAT pathway components on inducing dendrite hyperarborization in HYD are quite variable! Three Dome RNAi KDs had no effect and one stat92E KD had an effect while the other had not! What are the effects on Wg expressions in muscles with these genetic manipulations? Since wg transcription could be regulated in a complex way, the Dome/JAK/STAT pathway is likely insufficient to be the main or only regulator. This is also echoed by a result described in the Discussion that Stat92E was not elevated by VMC supplement in LYD. The authors should limit their description on the significance of these results such as the description in Abstract (and Figure 7) and conclusion on lines 276-279 in Results. Also, they could include more discussion on the possibility of other pathways.

5. (line 217-218) "However, these results need careful interpretation (see the Figure 4—figure supplement 2 legend)" From the legend "However, knocking down dsh or blocking *JNK* signaling tended to increase the branch density over the control genotype on HYD, which may consequently reduce the differences in densities of dendritic terminals between the diets." I do not quite understand the reasoning here why some increases in HYD in the KD may reduce the differences in LYD? The explanation in the Figure legend is not clear!

*Reviewer #3 (Recommendations for the authors):*

The Akt phenotype is quite different from the Ror one based on the images shown in Figure 2. It looks like it causes a baseline phenotype in HYD, and overall seems to affect more parameters of dendrite structure than Ror. It is therefore unlikely that Akt, which has a stronger and different phenotype from Ror, is the downstream mediator of Ror signaling. Indeed, as the authors mention in the discussion, Akt has previously been shown to regulate ddaC dendrite branching through signals from epithelial cells, and it has also been previously demonstrated, that, unlike Ror, Akt affects dendrite growth on standard diets (Parrish, 2009).

In this study in the supplemental data, as well as in a previous study (Poe, 2020) (that interprets the overgrowth of ddaC neurons on LYD in a somewhat different way) InR has been shown to be important. It is not clear why it is dismissed in this study when it is much better established as a regulator of Akt than Ror.

The fact that Ror and Akt are interpreted as having similar effects on dendrite growth makes me very concerned about the quantitation method used. It is somewhat opaque and also at times dismissed (for example the growth phenotypes of InR and Alk are described as difficult to interpret and so sidelined, but it is unclear why).

The only data that suggests that Ror functions upstream of Akt are the phosphor-Akt staining shown in Figure 4. The antibody used is different from one previously used in da neurons in Parrish 2009 and looks like it is from a commercial source. With the recognition that much of the reproducibility crisis in science is due to poorly validated antibodies, it is essential to include key controls to validate this antibody- including loss of signal when Akt is knocked down. The differences in signal do not look particularly robust, for example when compared to those shown in Parrish 2009, and as the entire link between Ror and Akt rests on this data, it is imperative to be very sure that it is correct, and ideally do some additional experiments to determine whether Ror actually acts through Akt.

I would also suggest considering an alternate model, which seems much more likely based on data in the supplement and previous data on Ror function in ddaC neurons. To promote branching during dendrite regeneration Ror functions through canonical Wnt signaling proteins including dsh and Axin (Nye, 2020), which these authors also show have branching phenotypes in their assay. This Ror function is part of a Wnt signaling pathway that controls microtubule nucleation and also includes arrow, fz, and fz2 (Nye, 2020 and Weiner, 2020). The failure to exhibit excess growth by Ror knockdown neurons in LYD seems likely to be related to the function of Ror in mediating microtubule nucleation in these cells.

It looks like the major change for wg RNAi in muscle is a change in arbor area in LYD – indeed this often seems to drive or at least contribute to the phenotype (except in Akt loss). Is this because animals are overall larger? Since the arbors cover the body – is this increase in size related to overall size? If so, is the dendrite difference secondary to a much broader change in animal size? Are the same conclusions reached if the total number of terminals is shown?

I do not understand why some of the results are dismissed, but others are not. For example: why do the dsh k/d and bskDN results in more careful interpretation than, say the fz2 or fz knockdown? The reason given is that there is a slight increase in branching in HYD, but it also looks like that might be the case in Ror mutants, while the opposite is seen in Ror k/d.

There issue of lengthened developmental time on LYD is problematic and not sufficiently addressed. In a previous study on this topic (Poe), the continued growth of ddaC neurons was interpreted as resilience to malnutrition and shown to be controlled by lower levels of a stress transcription factor, foxo, than other cells. In this interpretation, the ddaC neurons just continue to grow at their normal pace while the overall development of the animal is slowed. How is this view reconciled with the current model? Which conditions change developmental timing in this study? Do any of the genetic manipulations rescue developmental speed- and if so, is dendrite architecture changing secondarily to this?

Correct controls are needed for RNAi experiments. Control RNAis should be paired with other transgenes rather than no RNAis to control for Gal4 dilution by expression of multiple transgenes. For example, in Figure 4 myr-Akt is compared to Ror k/d + myr-Akt. Ror k/d + myr-Akt should be compared to control RNAi + myr-Akt.

In Figure 4Supp1 it states in the legend that both fz and fz2 knockdown were different from control, but only fz2 is mentioned in the main text. In this figure, images are shown for an fz2 mutant, but no quantitation is shown.

Figure 5 – data seems a little preliminary. It might be better to figure out the pathway in which Ror acts rather than add another piece.

In figure 6 – it would be helpful to include Ror k/d in the electrophysiological assays. The behavioral assays look somewhat inconclusive in terms of whether Ror rescues the LYD phenotype.

---

## [Author Response]

Essential revisions:One of the reviewers has consolidated and summarized our consultations on the individual reviews. These are given below. Please attend to each concern while responding. Please all see the individual reviews below.1) Are Wg levels increased in muscles when Upd2, Dome, or stat93E KD is knocked down. Since wg transcription could be regulated in a complex way, the Dome/JAK/STAT pathway is likely insufficient to be the main or only regulator. This is also echoed by a result described in the Discussion that Stat92E was not elevated by VMC supplement in LYD. The authors should limit their description on the significance of these results such as the description in Abstract (and Figure 7) and conclusion on lines 276-279 in Results. Also, they could include more discussion on the possibility of other pathways.

We agree with the reviewers that it is likely that signaling pathways other than the Dome/JAK/STAT could also contribute to downregulation of Wg on the high-yeast diet (HYD). This is because we conducted two experiments to address whether the Dome/JAK/STAT is a primary contributor to downregulating Wg on HYD, but the results were not supportive of such a possibility: First, when *stat92E* was knocked down in muscles, Wg increased on HYD; however, that increase was marginal (Figure 5F and 5G) compared to the change in the amount of Wg between larvae on HYD and on LYD (Figure 5F and 5H; see the quantitative data in 5J). Second, *stat92E* was not elevated by VMC supplementation on LYD (data not shown) as described in the Discussion in the original manuscript. Accordingly, we have followed the reviewers’ advice, carefully limiting our interpretation of the results, and we discuss the possible contribution of the hypothetical signaling pathways in the Abstract (lines 23-27), Introduction (lines 105-108), Results (lines 338-347) and Discussion (lines 441-448), as well as our model (Figure 7).

2) Could authors compare the KD efficiency in two Akt RNAi lines (v2902 and BL33615) (or are they known in previous literature)? They also need to test the specificity of anti-Akt antibodies using these akt RNAi lines. Are anti-Akt signals gone when Akt is knocked down?

The reviewers are requesting the KD efficiency in two *Akt* RNAi lines (v2902 and BL33615) and the specificity of the anti-phospho-Akt (Serine 473) antibody employed in our study (Cell Signaling). The KD efficiency in those RNAi lines and the specificity of the antibody have been documented in other developmental contexts in previously published literature. For example, when *Akt* is knocked down in the v2902 line, p-Akt signals are absent in the wing imaginal disc (see Figure 3E in Santabarbara-Ruiz et al. *PLoS Genetics*, 2019).

We have performed additional experiments and evaluated the KD efficiency in the lines and the specificity of the antibody in the class IV da neuron ddaC as follows:

1. p-Akt signals were significantly reduced in either the v2902 or BL33615 line (Figure 2—figure supplement 3A-3H).

2. The KD efficiency differed between the two lines, which is consistent with the phenotypic differences. The BL33615 line, which gave more severe dendrite phenotypes (Figure 2A-2J), showed weaker p-Akt signals than v2902 (Figure 2—figure supplement 3A-3H of the revised version).

3. Expression of myr-Akt, a constitutively activated membrane-anchored form of Akt (Stocker et al., *Science*, 2002) in class IV da neurons, increased the signal level of p-Akt (Figure 2—figure supplement 3I and 3J).

We added the above results in our revised manuscript (lines 175-176 and 268-272 in Results).

3) Was the dendrite hyper-arborization phenotype also observed in other types of da neurons? Did you see similar pAkt upregulation in other da neurons (class I-III) under the condition of LYD diets? If the authors have the results, it should be very nice to demonstrate the general and significant effects of their findings and mechanism.

The reviewer is asking whether the hyperarborization phenotype and p-Akt upregulation are also observed in other classes of da neurons. We addressed this question by performing additional quantitative data analyses in class I and III da neurons:

1. We previously reported that the hyperarborization phenotype is not seen in class I da neurons, ddaD and ddaE (Figure 2 in Watanabe et al., *Genes to Cells*, 2017). We quantified p-Akt levels in class I da neuron ddaE and found that the p-Akt level showed no significant difference between HYD and LYD (Figure 4—figure supplement 3A-3C in the revised manuscript).

2. On the other hand, class III da neuron ddaF showed increases both in the dendritic terminal number and in the p-Akt level on LYD compared to HYD (Figure 4—figure supplement 3D-3I), much like class IV da neurons.

Our results are consistent with a previous study (Poe et al., *eLife,* 2020), which examined dendrite morphologies (but not p-Akt signals), raising the possibility that the Akt-driven branching mechanism in response to the low-nutrient condition in class IV da neurons is shared by other classes of da neurons, including class III. We described and discussed these results in our revised manuscript (lines 288-300 in Results).

4) The Akt mutant phenotype is quite different from the Ror mutant phenotype based on the images shown in Figure 2. It seems to affect more parameters of dendrite structure than Ror. The authors should elaborate on the phenotypic difference between Akt and Ror mutants, perhaps using different quantification approaches. It is therefore appears unlikely that Akt, which has a stronger and different phenotype from Ror, only mediates Ror signaling. Indeed, as the authors mentioned in the Discussion, Akt has previously been shown to regulate ddaC dendrite branching through signals from epithelial cells, and it has also been previously demonstrated, that, unlike Ror, Akt affects dendrite growth on standard diets (Parrish, 2009). The only data that suggests that Ror functions upstream of Akt are the pAkt staining shown in Figure 4. The authors need to confirm the specificity of the Akt antibody. In addition, the authors should address this concern by modifying the model in the Discussion (such as to include the possibility of excess growth by Ror knockdown neurons in LYD seems likely to be related to the function of Ror in mediating microtubule nucleation, the possibility of integrating InR and AlK in the pathway, which they have shown in this study in the supplemental data, as well as in a previous study (Poe, 2020)).

The reviewers pointed out that the *Akt*-knocked down phenotype is quite different from the *Ror*-knocked down or mutant phenotype (Figure 2) and gave helpful suggestions. Before explaining our additional analysis on this matter, we addressed the specificity of the p-Akt antibody in class IV da neurons and the differential KD efficiency in two Akt RNAi lines in our response to Essential Revisions 2, and we have consolidated our model where Ror functions upstream of Akt (Figure 4).

To elaborate the phenotypic difference between *Akt* KD and *Ror* KD, we followed the reviewers’ advice and introduced a different quantification approach. In addition to dendrite “branching,” we evaluated “elongation” in our revised manuscript (Figure 2—figure supplement 2). We measured the total length of branches per neuron (dendrite length) and also divided the total length by the arbor size (dendrite length/area). Box plots of dendrite length/area (Figure 2—figure supplement 2B, 2D, and 2F) and that of terminal number/area (Figure 1E) showed obvious “hyperelongation” as well as “hyperarborization” of the control class IV da neuron on LYD compared to HYD. In addition to the box plots, we drew two-dimensional plots with the dendritic area on the X-axis and the dendrite length on the Y-axis (Figure 2—figure supplement 2A, 2C and 2E), which showed that the numerical features of dendrite length of the control neurons on HYD and those on LYD were clearly separated (red solid ellipse and blue solid ellipse, respectively).

Compared to the control neurons, strong *Akt* KD in the BL33615 line severely impaired both elongation (Figure 2—figure supplement 2C and 2D) and branching (Figure 2C, 2F, 2I and 2J), irrespective of the diets. On the other hand, *Akt* KD in the v2902 line mildly affected the hyperelongation phenotype (Figure 2—figure supplement 2A and 2B); and it did ameliorate hyperarborization on LYD (Figure 2B, 2E, 2G and 2H). These results indicate that the basal activity of Akt is required for both elongation and branching on HYD and LYD (Figure 2—figure supplement 2G-2I), which is consistent with a previous report that demonstrated the requirement of Akt for the regulation of dendritic morphology of class IV da neurons on standard laboratory food (Parrish et al., *Neuron,* 2009).

Our critical findings are: (1) *Ror* KD or a *Ror* mutation blunted both elongation and branching only on LYD (Figure 2—figure supplement 2E and 2F and Figure 2K-2V); and (2) the diet-dependent phenotype of the mild *Akt* KD by the v2902 line was partly similar to the *Ror* KD or mutant phenotype. Together with the evidence for the function of Ror upstream of Akt (Figure 4), our result reinforces the proposed role of the Ror/Akt signaling pathway in response to LYD. We have drawn diagrams of how the signaling pathway is working or defective and how it affects dendrite branching as well as elongation under individual genetic and dietary conditions (Figure 2—figure supplement 2G-2J). Our model underscores the notion that Akt mediates signaling from multiple upstream receptors, including Ror. We also revised the text throughout the manuscript (lines 169-180, 202-210, and 422-425) and Figure 7.

Regarding questions related to the function of Ror in mediating microtubule nucleation and the possibility of integrating InR and AlK in the pathway, please see our detailed responses to major points 2, 5, and 7 of reviewer #3.

5) It looks like the major change for wg RNAi in muscle is a change in arbor area in LYD- indeed this often seems to drive or at least contribute to the phenotype (except in Akt loss). Is this because animals are overall larger? Since the arbors cover the body- is this increase in size related to overall size? If so, is the dendrite difference secondary to a much broader change in animal size? Are the same conclusions reached if a total number of terminals is shown?

The reviewers are concerned about the possibility that the suppression of hyperarborization by *wg* knockdown in muscle may be a secondary effect of an increase in arbor size or larval body size (compare the blue solid ellipse with the blue dotted one in Figure 3E). We did not measure larval body size and cannot answer whether the *wg* knocked down larvae were bigger than control larvae on LYD. However, we believe that the suppression of hyperarborization by *wg* knockdown was not a secondary effect of the increased body size, for the following reasons:

1. As mentioned in the legend for Figure 1, 2D-plots of the control larvae, such as Figure 1F, show (1) a positive correlation between the area of the dendritic field and the number of branch terminals, and (2) a clear separation of 95% confidence ellipses of the numerical values between HYD and LYD (compare the blue ellipse with the red one). Therefore, instead of simply comparing the number of branch terminals between the diets, we focused on changes in the density of terminals (Figure 1E) and how far the pair of red and blue ellipses are separated in the 2D plot (Figure 1F) to evaluate the hyperarborization phenotypes. When *wg* was knocked down in muscles on LYD, its ellipse (the blue dotted one in Figure 3E) shifted closer to or overlapped with ellipses of the larvae on HYD (the red solid and dotted ones in Figure 3E). This result suggests that *wg* RNAi in muscles blunted the hyperarborization phenotype despite the increase in the dendritic area.

2. We also analyzed the effects of *wg* knockdown using another muscle GAL4 driver (Figure 3—figure supplement 1A-1F) and *wg* hypomorphic mutations (Figure 3G-3L). The distribution of the arbor size on LYD was not much affected, neither by the knockdown nor the mutation (compare blue solid circles with open circles in Figure 3—figure supplement 1E and Figure 3K); and again, the ellipse of the knockdown or the mutant larvae shifted closer to or overlapped with those of larvae on HYD (red ellipses in the 2D plots, respectively). Together with the box plots of the terminal number/area (Figure 3—figure supplements 1F and Figure 3L), our results strongly suggest that reduced function of *wg* attenuates the hyperarborization phenotype without causally changing the body size.

6) The issue of lengthened developmental time on LYD is problematic and not sufficiently addressed. In a previous study on this topic (Poe), the continued growth of ddaC neurons was interpreted as resilience to malnutrition and shown to be controlled by lower levels of a stress transcription factor, foxo, than in other cells. In this interpretation, the ddaC neurons just continue to grow at their normal pace while the overall development of the animal is slowed. How is this view reconciled with the current model? Which conditions change developmental timing in this study? Do any of the genetic manipulations rescue developmental speed- and if so, is dendrite architecture changing secondarily to this?

The reviewers are concerned about the effect of the lengthened developmental time on LYD on dendrite architecture. The critical question is whether the hyperaborization phenotype is a secondary consequence of the longer larval stage on LYD than that on HYD. We gathered three lines of evidence against this possibility, which consist of the data already shown in the original submission and new data from two additional experiments. We explain each in the details below and also in the revised manuscript.

1. As stated in the original manuscript, we collected wandering 3rd instar larvae and imaged class IV da neurons as scheduled throughout our dietary or genetic interventions (6-7 days after egg laying (AEL) on HYD and 9-10 days AEL on LYD or LYD plus supplements; Figure 1—figure supplement 1A). On LYD plus any combinations of nutrients, larval developmental timings were essentially the same as for LYD; nonetheless, the hyperarborization phenotype was blunted on LYD+VMC(O) (Figure 1G-1U). Moreover, throughout the testing of control genotypes and all genetic interventions, the timings on LYD were similarly longer than those on HYD; notwithstanding, *Ror* KD (Figure 2K-2P), a *Ror* mutation (Figure 2Q-2V), *wg* KD in muscles (Figure 3A-3F) or *wg* mutations (Figure 3G-3L) blunted the hyperarborization phenotype compared to that of the WT dendritic arbors. These results indicate that our dietary or genetic interventions ameliorated hyperarborization without changing the developmental timing on LYD.

2. As briefly stated in the Introduction in the original version of the manuscript, we previously compared dendrite morphologies between a low-sugar diet and a high-sugar diet (Musselman et al., *Dis Model Mech,* 2011), the latter of which delays larval development, and we reported that the hyperarborization is not observed between those diets (Watanabe et al., *Genes to Cells*, 2017). We have now expanded this approach and analyzed the effect of the sugar overload on the arborization in a quantitative manner (Figure 1—figure supplement 2A-2E). The larval stage was longer on HYD supplemented with excess sucrose (HYD + sucrose) than on HYD; however, the dendritic arbors of class IV da neurons did not become more complex (Figure 1—figure supplement 2C-2E). Thus, we showed that an increase in the amount of sucrose in HYD, which extended larval development, was not associated with hyperarborization.

3. Finally, we addressed whether any of the genetic manipulations that cause larval developmental delay are associated with an increase in dendrite complexity on a standard diet or not. Many genetic manipulations are reported to cause developmental delays; however, most of those also result in large increases in the body size (e.g., McBrayer et al., *Developmental Cell,* 2007), which may complicate the matters in solving our key question. We therefore chose *dlip8* expression in wing imaginal discs, which is sufficient to extend the larval stage with a minimum effect on body growth (Colombani et al., *Science* 2012). This genetic intervention caused a mild delay in larval development, but it was not associated with hyperarborization (Figure 1—figure supplement 2F-2J), providing an additional piece of evidence allaying the concern about the effect of the lengthened developmental time on dendrite complexity.

The reviewers may wish to propose an experiment in which a genetic manipulation rescues larval developmental delay on LYD, and then examine whether the hyperarborization remains or not. This would be an interesting--but complex—experiment, which would require comparisons between genotypes, diets, and timings. We believe that our data above sufficiently dismiss the simple possibility of the hyperaborization phenotype as a secondary consequence of the longer larval stage on LYD; thus, we propose our model wherein class IV da neurons are programmed to arborize in excess due to the indispensable role of Wg/Ror/Akt signaling in response to combined VMC deficiency. As discussed in the original manuscript and in the revised version, the Poe et al. paper highlights the relationship between the neuron and the adjacent epidermis, where the key molecular mechanism is the differential expression of the transcription factor FoxO. In contrast, our study unearths the communication between the neuron and another adjacent tissue, the body wall muscle, which upregulates Wg in response to low-nutrient conditions. Both mechanisms can coexist and contribute to the continued growth of ddaC neurons while the overall development of the animal is slowed.

7) In figure 6 – it would be helpful, if feasible speedily, to include Ror k/d in the electrophysiological assays. The behavioral assays look somewhat inconclusive in terms of whether Ror rescues the LYD phenotype.

We appreciate the reviewer’s advice. Unfortunately, it has been difficult to rapidly conduct electrophysiological assays using larvae of multiple genotypes on different diets, due to limited human resources and access to apparatuses that must be shared by many other projects in the lab. As discussed in the original submission and also in the revised version, the identification of the downstream circuits would allow further studies, including electrophysiological analysis along the circuit, but this is beyond the scope of the present study.

8) On page 9 the authors mentioned screening 20 RTKs. It would be helpful to include this data. Certainly, two others they include – InR and Alk – look like they have similar phenotypes to Ror and it would be helpful to know the specificity.

Thank you for the suggestion. Supplementary file 2 lists the names of the 20 RTK genes, stock numbers of RNAi lines, GAL4 drivers, and effects of individual RNAi lines. We conducted two rounds of screening. In the primary screening, we intended to enhance the knockdown efficacy and used *Gr28b.c-GAL4* and *ppk-GAL4* together. We acquired images of 3-8 knocked-down neurons for each gene on each diet, and then visually judged whether hyperarborization was blunted or not. We selected nine genes (*Ror*, *InR*, *Alk*, *htl*, *Egfr*, *Pvr*, *Ddr*, *dnt*, and *drl*) for the secondary screening, in which we used *ppk-GAL4* only, because *Gr28b.c-GAL4* is expressed in a small subset of neurons in the central nervous system in addition to class IV da neurons in the peripheral nervous system (Xiang et al., *Nature*, 2010). Screening the 20 RTK genes is described in a new section in Materials and Methods in the revised manuscript (lines 542-550).

Representative images for each gene knockdown are shown in Figure 2—figure supplement 4 (data for the primary screening of 18 genes), Figure 2—figure supplement 5 (data for the primary and secondary screenings of *InR* and *Alk* and data for the secondary screening of *htl*), and Figure 2—figure supplement 6 (data for the secondary screening of *Egfr*, *Pvr*, *Ddr*, *dnt*, and *drl*). Regarding the KD data for *InR* or *Alk*, see our explanations and interpretations in the response to point 2 of reviewer #3.

9) There are two ways of statistics when manipulating gene activities (RNAi or OE) in HYD and LYD. Direct comparisons of two samples were used mostly in the main Figures (except Figure 5Z), while "difference in two differences" were shown mainly for several Supplemental figures (2, 3, 4, 5). It is not clear why the authors used two different ways. Are there specific reasons for doing so? Could not the direct comparisons of two samples (such as for main figures) could cope with descriptions in the text and easier to comprehend.

The hyperarborization phenotype is defined based on the difference in the terminal density between the two diets (double-headed arrows in Author response image 1). In most analyses in this study, we examined which genetic manipulation blunted the hyperarborization. As rendered in Author response image 1, the gene KD in the lefthand plot did not blunt the hyperarborization phenotype, whereas the gene KD in the righthand plot did. To make a statistical decision, we employed 2-way ANOVA, which was also used in the previous study (Poe et al., *eLife*, 2020), to test the interaction between the two variables: genotype and diet. When we first introduce 2-way ANOVA in the revised manuscript (Figure 2H), we briefly explain why we needed to employ that particular statistical test (lines 162-168 in Results). The use of 2-way ANOVA is now also described in the Statistical Analysis section in Materials And Methods and in the figure legends.

On the other hand, when comparing the effects of genotypes under the same dietary conditions (Figures 3Y, 4V, 4W, 5S, 5T,5AF, and Figure 5—figure supplement2), we simply focused on whether a genetic manipulation itself affected dendrite branching. For example, in the *Mhc > wg* experiment (Figure. 3Y), we were interested in whether muscle-derived Wg promoted dendrite branching, not whether its effect varied according to the diets. We added this description to the Statistical Analysis section in Materials And Methods.

**Author response image 1. sa2fig1:** 

Reviewer #2 (Recommendations for the authors):1. Akt knockdowns in two lines (v2902 and BL33615) display qualitative suppression of dendrite hyperarborization in Figure 2. In v2902 KD, while the dendritic arbor looks more normal in both HYD and LYD, the BL33615 KD showed quite a dramatic reduction even in HYD, and also in LYD, rendering both sets of data groups clustering on a distinct area (Figure 2I), instead of in between two controls (HYD and LYD) such as in Figure 2G. While these two lines have been used before, I am wondering if the BL33615 has a stronger KD effect and if the basal Akt is necessary for a fundamental process in normal dendrite development (which is still preserved in the v2902 line), irrespective of nutrient condition. Could authors compare the KD efficiency in these two lines (or are they known in previous literature)?

We thank for the reviewer’s important suggestion and addressed this point as described in Essential revision 2：

The reviewers are requesting the KD efficiency in two *Akt* RNAi lines (v2902 and BL33615) and the specificity of the anti-phospho-Akt (Serine 473) antibody employed in our study (Cell Signaling). The KD efficiency in those RNAi lines and the specificity of the antibody have been documented in other developmental contexts in previously published literature. For example, when *Akt* is knocked down in the v2902 line, p-Akt signals are absent in the wing imaginal disc (see Figure 3E in Santabarbara-Ruiz et al. *PLoS Genetics*, 2019).

We have performed additional experiments and evaluated the KD efficiency in the lines and the specificity of the antibody in the class IV da neuron ddaC as follows:

1. p-Akt signals were significantly reduced in either the v2902 or BL33615 line (Figure 2—figure supplement 3A-3H).

2. The KD efficiency differed between the two lines, which is consistent with the phenotypic differences. The BL33615 line, which gave more severe dendrite phenotypes (Figure 2A-2J), showed weaker p-Akt signals than v2902 (Figure 2—figure supplement 3A-3H of the revised version).

3. Expression of myr-Akt, a constitutively activated membrane-anchored form of Akt (Stocker et al., *Science*, 2002) in class IV da neurons, increased the signal level of p-Akt (Figure 2—figure supplement 3I and 3J).

We added the above results in our revised manuscript (lines 175-176 and 268-272 in Results).

2. The wg signal derived from muscles is required for dendrite hyperarborization (Figure 3) is a very nice set of data, although the effect by overexpression of wg in muscle is limited in HYD (Figure 3Y). It could be due to further control such as from epidermal cells shown in Poe et al. (2020) eLife. The experiment was performed at 29C to increase Wg expression, and the dendrite morphology was not normal. I am not asking for more experiments, but the authors could describe/explain what is the results at 25C instead of using 29C for readers not familiar with the approach? Have the authors tried using 2 copies of UAS-wg to increase wg expression at 25C?

The reviewer is asking about the effect of *wg* overexpression in muscles on dendrite arborization and our experimental design (Figures 3T-3Y). In addition to the overexpression at 29°C (Figures 3T-3Y), we did perform the same experiment at 25°C and found that the dendrite density increased on HYD (Author response image 2 and 2B). We added this description in the revised Figure3 legend. We have not used 2 copies of *UAS-wg* to increase its expression.

As the reviewer pointed out, the increase in the dendrite density by *wg* overexpression on either diet was less dramatic compared to the difference between the diets in each genotype at 29°C (Figures 3T-3Y), which is now stated as such in the revised version (lines 1027-1029). However, we do not discuss how this occurs, because it could be due to a number of possibilities. For example, the *wg* overexpression might not override the regulation by the epidermis, as the reviewer implied. While it would be quite interesting to pursue, it is also beyond the scope of this study to unravel the mechanisms underlying the limited dendrite growth due to *wg* overexpression.

3. There are two ways of statistics used when manipulating gene activities (RNAi or OE) in HYD and LYD. Direct comparisons of two samples were used mostly in the main Figures (except Figure 5Z), while "difference in two differences" were shown mainly for several Supplemental figures (2, 3, 4, 5). It is not clear why the authors used two different ways. Are there specific reasons for doing so? I thought the direct comparisons of two samples (such as for main figures) could cope with descriptions in the text and be easier to comprehend.

We addressed this point as described in Essential revision 9:

The hyperarborization phenotype is defined based on the difference in the terminal density between the two diets (double-headed arrows in Author response image 1). In most analyses in this study, we examined which genetic manipulation blunted the hyperarborization. As rendered in Author response image 1, the gene KD in the lefthand plot did not blunt the hyperarborization phenotype, whereas the gene KD in the righthand plot did. To make a statistical decision, we employed 2-way ANOVA, which was also used in the previous study (Poe et al., *eLife*, 2020), to test the interaction between the two variables: genotype and diet. When we first introduce 2-way ANOVA in the revised manuscript (Figure 2H), we briefly explain why we needed to employ that particular statistical test (lines 162-168 in Results). The use of 2-way ANOVA is now also described in the Statistical Analysis section in Materials And Methods and in the figure legends.

On the other hand, when comparing the effects of genotypes under the same dietary conditions (Figures 3Y, 4V, 4W, 5S, 5T,5AF, and Figure 5—figure supplement2), we simply focused on whether a genetic manipulation itself affected dendrite branching. For example, in the *Mhc > wg* experiment (Figure 3Y), we were interested in whether muscle-derived Wg promoted dendrite branching, not whether its effect varied according to the diets. We added this description to the Statistical Analysis section in Materials And Methods.

4. The RNAi KD effects of Dome/JAK/STAT pathway components on inducing dendrite hyperarborization in HYD are quite variable! Three Dome RNAi KDs had no effect and one stat92E KD had an effect while the other had not! What are the effects on Wg expressions in muscles with these genetic manipulations? Since wg transcription could be regulated in a complex way, the Dome/JAK/STAT pathway is likely insufficient to be the main or only regulator. This is also echoed by a result described in the Discussion that Stat92E was not elevated by VMC supplement in LYD. The authors should limit their description on the significance of these results such as the description in Abstract (and Figure 7) and conclusion on lines 276-279 in Results. Also, they could include more discussion on the possibility of other pathways.

We described our response to this point in detail in Essential revision 1 and clarified the limited contribution of the JAK/STAT pathway in the revised version as the reviewer suggested.

The reviewer stated “The RNAi KD effects of Dome/JAK/STAT pathway components on inducing dendrite hyperarborization in HYD are quite variable! Three Dome RNAi KDs had no effect and one stat92E KD had an effect while the other had not.” This statement may stem from an unfortunate misunderstanding of our data. *Stat92E* knockdown in muscles in both of the two RNAi lines similarly resulted in enhanced dendritic branching on HYD (Figure 5L, 5O, 5Q, and 5S, and Figure 5—figure supplement 2B, 2G, 2K and 2O). These results are consistent with our hypothesis that Stat92E negatively regulates *wg* in muscles, and consequently, branching of class IV da neurons on HYD. This enhanced branching was also seen with the knockdown of *dome* in one of the three RNAi lines (Figure 5—figure supplement 2C, 2H, 2L, and 2P), knockdown of *hop* (Figure 5M, P, R, and T), and knockdown of *upd2* in both of the two RNA lines (Figures 5AB, 5AD, 5AE, and 5AF, and Figure 5—figure supplement 2V, 2AB, 2AG and 2AL). Therefore, it seems reasonable to speculate that the Upd2- Stat92E pathway functions in suppressing the hyperarborization on HYD. However, we have no data to explain why only one out of the three *dome* RNAi lines promoted dendritic branching on HYD; thus, it will be necessary to verify the involvement of Dome in the future (lines 334-336 of the revised version).

5. (line 217-218) "However, these results need careful interpretation (see the Figure 4—figure supplement 2 legend)" From the legend "However, knocking down dsh or blocking JNK signaling tended to increase the branch density over the control genotype on HYD, which may consequently reduce the differences in densities of dendritic terminals between the diets." I do not quite understand the reasoning here why some increases in HYD in the KD may reduce the differences in LYD? The explanation in the Figure legend is not clear!

We addressed this concern in the revised manuscript, and offer our explanation, referring to Figure 4—figure supplement 2, as follows:

In our studies, we focused on the difference in dendrite terminal density between the two diets (HYD and LYD) and pursued dietary and genetic interventions that blunted this hyperarborization (see Author response image 1 in Essential revision 9). Reduction or loss of *Ror* function attenuated dendrite branching on LYD, while there were marginal effects on HYD (Figures 2K-2V in the revised manuscript). On the other hand, *dsh* knockdown or expression of a dominant-negative form of Bsk not only reduced dendrite branching on LYD but also increased dendritic branching on HYD (Figure 4-supplement 2B-2C, 2I-2J, 2O-2P and 2U-2V in the revised manuscript). Given this phenotypic difference on HYD between *Ror* and *dsh* or *bsk*, we speculate that Dsh and Bsk may regulate dendrite branching through a different mechanism than the low-nutrient dependent Wg-Ror pathway. We therefore did not focus on these factors further in this study.

Nonetheless, it is possible that these factors contribute to the hyperarborization phenotype either independently of the Wg-Ror-Akt pathway or by acting on Akt in a different manner from Wg-Ror signaling. Considering this possibility, we added a description to the Discussion and modified Figure 7.

Reviewer #3 (Recommendations for the authors):The Akt phenotype is quite different from the Ror one based on the images shown in Figure 2. It looks like it causes a baseline phenotype in HYD, and overall seems to affect more parameters of dendrite structure than Ror. It is therefore unlikely that Akt, which has a stronger and different phenotype from Ror, is the downstream mediator of Ror signaling. Indeed, as the authors mention in the discussion, Akt has previously been shown to regulate ddaC dendrite branching through signals from epithelial cells, and it has also been previously demonstrated, that, unlike Ror, Akt affects dendrite growth on standard diets (Parrish, 2009).

We thank the reviewer’s insightful suggestion. We addressed this point as described in Essential revision 4:

The reviewers pointed out that the *Akt*-knocked down phenotype is quite different from the *Ror*-knocked down or mutant phenotype (Figure 2) and gave helpful suggestions. Before explaining our additional analysis on this matter, we addressed the specificity of the p-Akt antibody in class IV da neurons and the differential KD efficiency in two Akt RNAi lines in our response to Essential Revisions 2, and we have consolidated our model where Ror functions upstream of Akt (Figure 4).

To elaborate the phenotypic difference between *Akt* KD and *Ror* KD, we followed the reviewers’ advice and introduced a different quantification approach. In addition to dendrite “branching,” we evaluated “elongation” in our revised manuscript (Figure 2—figure supplement 2). We measured the total length of branches per neuron (dendrite length) and also divided the total length by the arbor size (dendrite length/area). Box plots of dendrite length/area (Figure 2—figure supplement 2B, 2D, and 2F) and that of terminal number/area (Figure 1E) showed obvious “hyperelongation” as well as “hyperarborization” of the control class IV da neuron on LYD compared to HYD. In addition to the box plots, we drew two-dimensional plots with the dendritic area on the X-axis and the dendrite length on the Y-axis (Figure 2—figure supplement 2A, 2C and 2E), which showed that the numerical features of dendrite length of the control neurons on HYD and those on LYD were clearly separated (red solid ellipse and blue solid ellipse, respectively).

Compared to the control neurons, strong *Akt* KD in the BL33615 line severely impaired both elongation (Figure 2—figure supplement 2C and 2D) and branching (Figure 2C, 2F, 2I and 2J), irrespective of the diets. On the other hand, *Akt* KD in the v2902 line mildly affected the hyperelongation phenotype (Figure 2—figure supplement 2A and 2B); and it did ameliorate hyperarborization on LYD (Figure 2B, 2E, 2G and 2H). These results indicate that the basal activity of Akt is required for both elongation and branching on HYD and LYD (Figure 2—figure supplement 2G-2I), which is consistent with a previous report that demonstrated the requirement of Akt for the regulation of dendritic morphology of class IV da neurons on standard laboratory food (Parrish et al., *Neuron,* 2009).

Our critical findings are: (1) *Ror* KD or a *Ror* mutation blunted both elongation and branching only on LYD (Figure 2—figure supplement 2E and 2F and Figure 2K-2V); and (2) the diet-dependent phenotype of the mild *Akt* KD by the v2902 line was partly similar to the *Ror* KD or mutant phenotype. Together with the evidence for the function of Ror upstream of Akt (Figure 4), our result reinforces the proposed role of the Ror/Akt signaling pathway in response to LYD. We have drawn diagrams of how the signaling pathway is working or defective and how it affects dendrite branching as well as elongation under individual genetic and dietary conditions (Figure 2—figure supplement 2G-2J). Our model underscores the notion that Akt mediates signaling from multiple upstream receptors, including Ror. We also revised the text throughout the manuscript (lines 169-180, 202-210, and 422-425) and Figure 7.

In this study in the supplemental data, as well as in a previous study (Poe, 2020) (that interprets the overgrowth of ddaC neurons on LYD in a somewhat different way) InR has been shown to be important. It is not clear why it is dismissed in this study when it is much better established as a regulator of Akt than Ror.

For the following reasons, we focused on Ror, rather than InR or Alk, in this study. We have added a new Supplementary file 2 and Figure 2—figure supplement 5.

1) As for InR, while knockdown of *InR* by one RNAi line, which was used in Poe et al., attenuated the hyperarborization phenotype (Figure 2—figure supplement 5G-5L), another RNAi line (not used in Poe et al.) had no significant effect on the phenotype (Figure 2—figure supplement 5A-5F). Therefore, we could not convincingly conclude that InR contributes to the low-nutrient dependent hyperarborization.

2) *Alk* knockdown in the primary screening (using *ppk-GAL4 + Gr28b.c-GAL4*) resulted in blunted hyperarborization. However, knockdown using 2 RNAi lines in the secondary screening (using *ppk-GAL4*) had no significant effect (Supplementary file 2 and Figure 2—figure supplement 5M-5X).

3) On the other hand, *Ror* knockdown consistently suppressed the hyperarborization phenotype in both the primary and the secondary screening (Figure 2—figure supplement 4C and 4D and Figures 2K-2P). Moreover, mutant analyses recapitulated the knockdown results (Figures 2Q-2V). We therefore decided to investigate the Ror-mediated mechanism.

Future studies will explore whether other RTKs including InR and Alk indeed function upstream of Akt in the context of the Wg/Ror/Akt signaling, and if so, how these various inputs are integrated by Akt. Accordingly, we modified the Result (lines 198-201), Discussion (lines 422-425) and our model (Figure 7).

The fact that Ror and Akt are interpreted as having similar effects on dendrite growth makes me very concerned about the quantitation method used. It is somewhat opaque and also at times dismissed (for example the growth phenotypes of InR and Alk are described as difficult to interpret and so sidelined, but it is unclear why).

To elaborate the phenotypic differences and similarities between the *Ror* KD and the *Akt* KD, we followed the reviewers’ advice (introduction of different quantification approaches in Essential revision 4) and evaluated dendrite “elongation” in addition to “branching”. As described in detail in our reply to Essential revision 4, this quantitative analysis revealed the phenotypic similarity between the *Ror* KD and the *Akt* mild KD.

The only data that suggests that Ror functions upstream of Akt are the phosphor-Akt staining shown in Figure 4. The antibody used is different from one previously used in da neurons in Parrish 2009 and looks like it is from a commercial source. With the recognition that much of the reproducibility crisis in science is due to poorly validated antibodies, it is essential to include key controls to validate this antibody- including loss of signal when Akt is knocked down. The differences in signal do not look particularly robust, for example when compared to those shown in Parrish 2009, and as the entire link between Ror and Akt rests on this data, it is imperative to be very sure that it is correct, and ideally do some additional experiments to determine whether Ror actually acts through Akt.

The antibody used in Parrish et al. (2009) was no longer commercially available, and we had no choice but to use a different product. We opted to use the anti-phospho-Akt antibody (Cell Signaling) because it was used in a previous study that showed clear reduction of the signal in the wing imaginal disc when *Akt* was knocked down by *ptc-GAL4* (see Figure 3E in Santabárbara-Ruiz et al., *PLOS Genetics*, 2019). As described in detail in Essential revisions 2, we validated the specificity of the Cell Signaling antibody in class IV da neuron, ddaC. We believe that our immunostaining data for *Ror* knocked down neurons (Figures 4A-4E) and *wg* overexpressing larvae (Figures 4F-4J) convincingly indicate that Wg-Ror signaling regulates Akt activity in class IV da neurons.

I would also suggest considering an alternate model, which seems much more likely based on data in the supplement and previous data on Ror function in ddaC neurons. To promote branching during dendrite regeneration Ror functions through canonical Wnt signaling proteins including dsh and Axin (Nye, 2020), which these authors also show have branching phenotypes in their assay. This Ror function is part of a Wnt signaling pathway that controls microtubule nucleation and also includes arrow, fz, and fz2 (Nye, 2020 and Weiner, 2020). The failure to exhibit excess growth by Ror knockdown neurons in LYD seems likely to be related to the function of Ror in mediating microtubule nucleation in these cells.

The reviewer suggests that the Ror-mediated microtubule nucleation mechanism (Nye et al. 2020) may contribute to the hyperarborization phenotype. This is because Knockdown of *dsh* or *Axin* appeared to blunt hyperarborization (Figure 4—figure supplement 2B, 2I, 2O, 2U for *dsh* and 2G, 2N, 2T, 2Z for *Axin*). However, we found a phenotypic difference between *Ror* knockdown and the knockdown of *dsh* or Axin, and consider that Dsh and Axin might regulate dendrite branching through a mechanism different from the low-nutrient dependent Wg-Ror pathway (see our reply to comment 7 below). We added this point to the Discussion (lines 415-418 in DISCUSSION).

It looks like the major change for wg RNAi in muscle is a change in arbor area in LYD – indeed this often seems to drive or at least contribute to the phenotype (except in Akt loss). Is this because animals are overall larger? Since the arbors cover the body – is this increase in size related to overall size? If so, is the dendrite difference secondary to a much broader change in animal size? Are the same conclusions reached if the total number of terminals is shown?

We addressed this point as described in Essential revision 5:

The reviewers are concerned about the possibility that the suppression of hyperarborization by *wg* knockdown in muscle may be a secondary effect of an increase in arbor size or larval body size (compare the blue solid ellipse with the blue dotted one in Figure 3E). We did not measure larval body size and cannot answer whether the *wg* knocked down larvae were bigger than control larvae on LYD. However, we believe that the suppression of hyperarborization by *wg* knockdown was not a secondary effect of the increased body size, for the following reasons:

1. As mentioned in the legend for Figure 1, 2D-plots of the control larvae, such as Figure 1F, show (1) a positive correlation between the area of the dendritic field and the number of branch terminals, and (2) a clear separation of 95% confidence ellipses of the numerical values between HYD and LYD (compare the blue ellipse with the red one). Therefore, instead of simply comparing the number of branch terminals between the diets, we focused on changes in the density of terminals (Figure 1E) and how far the pair of red and blue ellipses are separated in the 2D plot (Figure 1F) to evaluate the hyperarborization phenotypes. When *wg* was knocked down in muscles on LYD, its ellipse (the blue dotted one in Figure 3E) shifted closer to or overlapped with ellipses of the larvae on HYD (the red solid and dotted ones in Figure 3E). This result suggests that *wg* RNAi in muscles blunted the hyperarborization phenotype despite the increase in the dendritic area.

2. We also analyzed the effects of *wg* knockdown using another muscle GAL4 driver (Figure 3—figure supplement 1A-1F) and *wg* hypomorphic mutations (Figure 3G-3L). The distribution of the arbor size on LYD was not much affected, neither by the knockdown nor the mutation (compare blue solid circles with open circles in Figure 3—figure supplement 1E and Figure 3K); and again, the ellipse of the knockdown or the mutant larvae shifted closer to or overlapped with those of larvae on HYD (red ellipses in the above 2D plots, respectively). Together with the box plots of the terminal number/area (Figure 3—figure supplements 1F and Figure 3L), our results strongly suggest that reduced function of *wg* attenuates the hyperarborization phenotype without causally changing the body size.

I do not understand why some of the results are dismissed, but others are not. For example: why do the dsh k/d and bskDN results in more careful interpretation than, say the fz2 or fz knockdown? The reason given is that there is a slight increase in branching in HYD, but it also looks like that might be the case in Ror mutants, while the opposite is seen in Ror k/d.

To address this concern, we added a brief description in the Results (lines 254-260), and we provide an explanation below in the figure legend (lines 1304-1311):

In our studies, we focused on the difference in dendrite terminal density between the two diets (HYD and LYD) and pursued dietary and genetic interventions that blunted this hyperarborization (see Author response image 1 in Essential revision 9). Reduction or loss of *Ror* function attenuated dendrite branching on LYD, while there were marginal effects on HYD (Figure 2K-2V in the revised manuscript). On the other hand, *dsh* knockdown or expression of a dominant-negative form of Bsk not only reduced dendrite branching on LYD but also increased dendritic branching on HYD (Figure 4-supplement 2B-2C, 2I-2J, 2O-2P and 2U-2V in the revised manuscript). This also appears to be the case with the *Axin* knockdown, although the effect was not statistically significant (Figure 4—figure supplement 2G, 2N, 2T and 2Z). Given this phenotypic difference on HYD between *Ror* and *dsh*, *bsk*, or *Axin*, we speculate that Dsh, Bsk and Axin might regulate the dendrite branching through a mechanism different from the low-nutrient dependent Wg-Ror pathway. We therefore did not focus on these factors further in this study.

There issue of lengthened developmental time on LYD is problematic and not sufficiently addressed. In a previous study on this topic (Poe), the continued growth of ddaC neurons was interpreted as resilience to malnutrition and shown to be controlled by lower levels of a stress transcription factor, foxo, than other cells. In this interpretation, the ddaC neurons just continue to grow at their normal pace while the overall development of the animal is slowed. How is this view reconciled with the current model? Which conditions change developmental timing in this study? Do any of the genetic manipulations rescue developmental speed- and if so, is dendrite architecture changing secondarily to this?

We understood that this reviewer’s concern is very important, performed two additional experiments, and described our response in Essential revision 6:

The reviewers are concerned about the effect of the lengthened developmental time on LYD on dendrite architecture. The critical question is whether the hyperaborization phenotype is a secondary consequence of the longer larval stage on LYD than that on HYD. We gathered three lines of evidence against this possibility, which consist of the data already shown in the original submission and new data from two additional experiments. We explain each in the details below and also in the revised manuscript.

1. As stated in the original manuscript, we collected wandering 3rd instar larvae and imaged class IV da neurons as scheduled throughout our dietary or genetic interventions (6-7 days after egg laying (AEL) on HYD and 9-10 days AEL on LYD or LYD plus supplements; Figure 1—figure supplement 1A). On LYD plus any combinations of nutrients, larval developmental timings were essentially the same as for LYD; nonetheless, the hyperarborization phenotype was blunted on LYD+VMC(O) (Figure 1G-1U). Moreover, throughout the testing of control genotypes and all genetic interventions, the timings on LYD were similarly longer than those on HYD; notwithstanding, *Ror* KD (Figure 2K-2P), a *Ror* mutation (Figure 2Q-2V), *wg* KD in muscles (Figure 3A-3F) or *wg* mutations (Figure 3G-3L) blunted the hyperarborization phenotype compared to that of the WT dendritic arbors. These results indicate that our dietary or genetic interventions ameliorated hyperarborization without changing the developmental timing on LYD.

2. As briefly stated in the Introduction in the original version of the manuscript, we previously compared dendrite morphologies between a low-sugar diet and a high-sugar diet (Musselman et al., *Dis Model Mech,* 2011), the latter of which delays larval development, and we reported that the hyperarborization is not observed between those diets (Watanabe et al., *Genes to Cells*, 2017). We have now expanded this approach and analyzed the effect of the sugar overload on the arborization in a quantitative manner (Figure 1—figure supplement 2A-2E). The larval stage was longer on HYD supplemented with excess sucrose (HYD + sucrose) than on HYD; however, the dendrites of class IV da neurons did not become more complex (Figure 1—figure supplement 2C-2E). Thus, we showed that an increase in the amount of sucrose in HYD, which extended larval development, was not associated with hyperarborization.

3. Finally, we addressed whether any of the genetic manipulations that cause larval developmental delay are associated with an increase in dendrite complexity on a standard diet or not. Many genetic manipulations are reported to cause developmental delays; however, most of those also result in large increases in the body size (e.g., McBrayer et al., *Developmental Cell,* 2007), which may complicate the matters in solving our key question. We therefore chose *dlip8* expression in wing imaginal discs, which is sufficient to extend the larval stage with a minimum effect on body growth (Colombani et al., *Science* 2012). This genetic intervention caused a mild delay in larval development, but it was not associated with hyperarborization (Figure 1—figure supplement 2F-2J), providing an additional piece of evidence allaying the concern about the effect of the lengthened developmental time on dendrite complexity.

The reviewers may wish to propose an experiment in which a genetic manipulation rescues larval developmental delay on LYD, and then examine whether the hyperarborization remains or not. This would be an interesting--but complex—experiment, which would require comparisons between genotypes, diets, and timings. We believe that our data above sufficiently dismiss the simple possibility of the hyperaborization phenotype as a secondary consequence of the longer larval stage on LYD; thus, we propose our model wherein class IV da neurons are programmed to arborize in excess due to the indispensable role of Wg/Ror/Akt signaling in response to combined VMC deficiency. As discussed in the original manuscript and in the revised version, the Poe et al. paper highlights the relationship between the neuron and the adjacent epidermis, where the key molecular mechanism is the differential expression of the transcription factor FoxO. In contrast, our study unearths the communication between the neuron and another adjacent tissue, the body wall muscle, which upregulates Wg in response to low-nutrient conditions. Both mechanisms can coexist and contribute to the continued growth of ddaC neurons while the overall development of the animal is slowed.

Correct controls are needed for RNAi experiments. Control RNAis should be paired with other transgenes rather than no RNAis to control for Gal4 dilution by expression of multiple transgenes. For example, in Figure 4 myr-Akt is compared to Ror k/d + myr-Akt. Ror k/d + myr-Akt should be compared to control RNAi + myr-Akt.

We have followed the reviewer’s advice, examined the effect of myr-Akt (the constitutively activated membrane-anchored form of Akt) using correct controls, and totally replaced the previous data (Figure 4K-4W of the original version) with the new data (Figure 4K-4W of the revised version). Fortunately, the results in the original version were recapitulated: Expression of myr-Akt in class IV da neurons increased the terminal density, regardless of whether *Ror* was knocked down or not. This result strengthened the idea that Akt activation in the neurons plays a pivotal role for the hyperarborization. Our choices of the controls are described in the legend, and the exact genotypes are listed in Supplementary file 4.

In Figure 4Supp1 it states in the legend that both fz and fz2 knockdown were different from control, but only fz2 is mentioned in the main text. In this figure, images are shown for an fz2 mutant, but no quantitation is shown.

We added our interpretation of the *fz* KD experiment to the legend of Figure 4—figure supplement 1 as follows:

Both 2D-plots and boxplots of dendrite density show that the degree of attenuation of dendrite branching on LYD is much higher in *fz2* KD neurons than *fz* KD neurons (Figure 4—figure supplement 1A-1J). Therefore, we mentioned only the result of *fz2* knockdown in the main text. However, we do not rule out the possibility that Fz may be involved in the hyperarborization phenotype.

We added the quantification data for the *fz2* mutant in Figure 4—figure supplement 1O and 1P and showed that hyperarborization was strongly suppressed in the *fz2* mutant neurons.

Figure 5 – data seems a little preliminary. It might be better to figure out the pathway in which Ror acts rather than add another piece.

We believe that the addition of the Dome/JAK/STAT pathway is informative because of its connection with *wg* expression in muscles. As described in Essential revision 1, we carefully clarified the limited contribution of the Dome/JAK/STAT pathway in the revised version.

In figure 6 – it would be helpful to include Ror k/d in the electrophysiological assays. The behavioral assays look somewhat inconclusive in terms of whether Ror rescues the LYD phenotype.

We replied to this suggestion in Essential revision 7:

We appreciate the reviewer’s advice. Unfortunately, it has been difficult to rapidly conduct electrophysiological assays using larvae of multiple genotypes on different diets, due to limited human resources and access to apparatuses that must be shared by many other projects in the lab. As discussed in the original submission and also in the revised version, the identification of the downstream circuits would allow further studies, including electrophysiological analysis along the circuit, but this is beyond the scope of the present study.

Reference

Colombani J, Andersen DS, Léopold P. 2012. Secreted Peptide Dilp8 Coordinates *Drosophila* Tissue Growth with Developmental Timing. *Science (1979)* 336:582–585. doi:10.1126/science.1216689

McBrayer Z, Ono H, Shimell MJ, Parvy JP, Beckstead RB, Warren JT, Thummel CS, Dauphin-Villemant C, Gilbert LI, O’Connor MB. 2007. Prothoracicotropic Hormone Regulates Developmental Timing and Body Size in *Drosophila*. *Dev Cell* 13:857–871. doi:10.1016/j.devcel.2007.11.003

Nye DMR, Albertson RM, Weiner AT, Ian Hertzler J, Shorey M, Goberdhan DCI, Wilson C, Janes KA, Rolls MM. 2020. The receptor tyrosine kinase Ror is required for dendrite regeneration in *Drosophila* neurons. *PLoS Biol* 18. doi:10.1371/journal.pbio.3000657

Palanker Musselman L, Fink JL, Narzinski K, Ramachandran PV, Sukumar Hathiramani S, Cagan RL, Baranski TJ. 2011. A high-sugar diet produces obesity and insulin resistance in wild-type *Drosophila*. *Dis Model Mech* 4:842–849. doi:10.1242/dmm.007948

Parrish JZ, Xu P, Kim CC, Jan LY, Jan YN. 2009. The microRNA bantam Functions in Epithelial Cells to Regulate Scaling Growth of Dendrite Arbors in *Drosophila* Sensory Neurons. *Neuron* 63:788–802. doi:10.1016/j.neuron.2009.08.006

Poe AR, Xu Y, Zhang C, Lei J, Li K, Labib D, Han C. 2020. Low FoxO expression in *Drosophila* somatosensory neurons protects dendrite growth under nutrient restriction. *eLife* 9:1–47. doi:10.7554/*eLife*.53351

Santabárbara-Ruiz P, Esteban-Collado J, Pérez L, Viola G, Abril JF, Milán M, Corominas M, Serras F. 2019. Ask1 and Akt act synergistically to promote ROS-dependent regeneration in *Drosophila*. *PLoS Genet* 15:e1007926. doi:10.1371/journal.pgen.1007926

Stocker H, Andjelkovic M, Oldham S, Laffargue M, Wymann MP, Hemmings BA, Hafen E. 2002. Living with Lethal PIP3 Levels: Viability of Flies Lacking PTEN Restored by a PH Domain Mutation in Akt/PKB. *Science (1979)* 295:2088–2091. doi:10.1126/science.1068094

Watanabe K, Furumizo Y, Usui T, Hattori Y, Uemura T. 2017. Nutrient-dependent increased dendritic arborization of somatosensory neurons. *Genes to Cells* 22:105–114. doi:10.1111/gtc.12451

Xiang Y, Yuan Q, Vogt N, Looger LL, Jan LY, Jan YN. 2010. Light-avoidance-mediating photoreceptors tile the *Drosophila* larval body wall. *Nature* 468:921–926. doi:10.1038/nature09576